# Coresets for Vertical Federated Learning: Regularized Linear Regression and $K$-Means Clustering

**Lingxiao Huang**[*]
Nanjing University
huanglingxiao1990@126.com

**Zhize Li**[*]
Carnegie Mellon University
zhizeli@cmu.edu

**Jialin Sun**[*]
Fudan University
sunjl20@fudan.edu.cn

**Haoyu Zhao**[*]
Princeton Univeristy
haoyu@princeton.edu

## Abstract

Vertical federated learning (VFL), where data features are stored in multiple parties distributively, is an important area in machine learning. However, the communication complexity for VFL is typically very high. In this paper, we propose a unified framework by constructing *coresets* in a distributed fashion for communication-efficient VFL. We study two important learning tasks in the VFL setting: regularized linear regression and $k$-means clustering, and apply our coreset framework to both problems. We theoretically show that using coresets can drastically alleviate the communication complexity, while nearly maintain the solution quality. Numerical experiments are conducted to corroborate our theoretical findings.

## 1 Introduction

Federated learning (FL) [54, 40, 44, 36, 68] is a learning framework where multiple clients/parties collaboratively train a machine learning model under the coordination of a central server without exposing their raw data (i.e., each party's raw data is stored locally and not transferred). There are two large categories of FL, horizontal federated learning (HFL) and vertical federated learning (VFL), based on the distribution characteristics of the data. In HFL, different parties usually hold different datasets but all datasets share the same features; while in VFL, all parties use the same dataset but different parties hold different subsets of the features (see Figure 1a).

Compared to HFL, VFL [74, 50, 71] is generally harder and requires more communication: as a single party cannot observe the full features, it requires communication with other parties to compute the loss and the gradient of *a single data*. This will result in two potential problems: (i) it may require a huge amount of communication to jointly train the machine learning model when the dataset is large; and (ii) the procedure of VFL transfers the information of local data and may cause privacy leakage. Most of the VFL literature focus on the privacy issue, and designing secure training procedure for different machine learning models in the VFL setting [28, 74, 72, 10]. However, the communication efficiency of the training procedure in VFL is somewhat underexplored. For unsupervised clustering problems, Ding et al. [19] propose constant approximation schemes for $k$-means clustering, and their communication complexity is *linear* in terms of the dataset size. For linear regression, although the communciaiton complexity can be improved to *sublinear* via sampling, such as SGD-type uniform sampling for the dataset [50, 74], the final performance is not comparable to that using the whole

---

[*]Alphabetical order.

dataset. Thus previous algorithms usually do not scale or perform well to the big data scenarios. [2] This leads us to consider the following question:

> *How to train machine learning models using sublinear communication complexity in terms of the dataset size without sacrificing the performance in the vertical federated learning (VFL) setting?*

In this paper, we try to answer this question, and our method is based on the notion of *coreset* [27, 22, 23]. Roughly speaking, coreset can be viewed as a small data summary of the original dataset, and the machine learning model trained on the coreset performs similarly to the model trained using the full dataset. Therefore, as long as we can obtain a coreset in the VFL setting in a communication-efficient way, we can then run existing algorithms on the coreset instead of the full dataset.

**Our contribution**   We study the communication-efficient methods for vertical federated learning with an emphasis on scalability, and design a general paradigm through the lens of coreset. Concretely, we have the following key contributions:

1. We design a unified framework for coreset construction in the vertical federated learning setting (Section 3), which can help reduce the communication complexity (Theorem 2.5).

2. We study the regularized linear regression (Definition 2.1) and $k$-means (Definition 2.2) problems in the VFL setting, and apply our unified coreset construction framework to them. We show that we can get $\varepsilon$-approximation for these two problems using only $o(n)$ sublinear communications under mild conditions, where $n$ is the size of the dataset (Section 4 and 5).

3. We conduct numerical experiments to validate our theoretical results. Our numerical experiments corroborate our findings that using coresets can drastically reduce the communication complexity, while maintaining the quality of the solution (Section 6). Moreover, compared to uniform sampling, applying our coresets can achieve a better solution with the same or smaller communication complexity.

## 1.1   More related works

**Federated learning**   Federated learning was introduced by McMahan et al. [54], and received increasing attention in recent years. There exist many works studied in the horizontal federated learning (HFL) setting, such as algorithms with multiple local update steps [54, 17, 39, 25, 56, 77] . There are also many algorithms with communication compression [38, 55, 47, 45, 24, 46, 60, 21, 76, 61, 78] and algorithms with privacy preserving [70, 30, 79, 64, 48].

**Vertical federated learning**   Due to the difficulties of VFL, people designed VFL algorithms for some particular machine learning models, including linear regression [50, 74], logistic regression [75, 73, 29], gradient boosting trees [63, 11, 10], and $k$-means [19]. For $k$-means, Ding et al. [19] proposed an algorithm that computes the global centers based on the product of local centers, which requires $O(nT)$ communication complexity. For linear regression, Liu et al. [50] and Yang et al. [74] used uniform sampling to get unbiased gradient estimation and improved the communication efficiency, but the performance may not be good compared to that without sampling. Yang et al. [73] also applied uniform sampling to quasi-Newton algorithm and improved communication complexity for logistic regression. People also studied other settings in VFL, e.g., how to align the data among different parties [62], how to adopt asynchronous training [9, 26], and how to defend against attacks in VFL [49, 53]. In this work, we aim to develop communication-efficient algorithms to handle large-scale VFL problems.

**Coreset**   Coresets have been applied to a large family of problems in machine learning and statistics, including clustering [22, 7, 31, 15, 16], regression [20, 43, 6, 13, 34, 12], low rank approximation [14], and mixture model [52, 33]. Specifically, Chhaya et al. [12] investigated coreset construction for regularized regression with different norms. Feldman and Langberg [22], Braverman et al. [7] proposed an importance sampling framework for coreset construction for clustering (including $k$-means). The coreset size for $k$-means clustering has been improved by several following works [31, 15, 16] to $\tilde{O}(k\varepsilon^{-4})$, and Cohen-Addad et al. [16] proved a lower bound of size $\Omega(\varepsilon^{-2}k)$. Due to the mergable

---

[2]In our numerical experiments (Section 6), we provide some results to justify this claim.

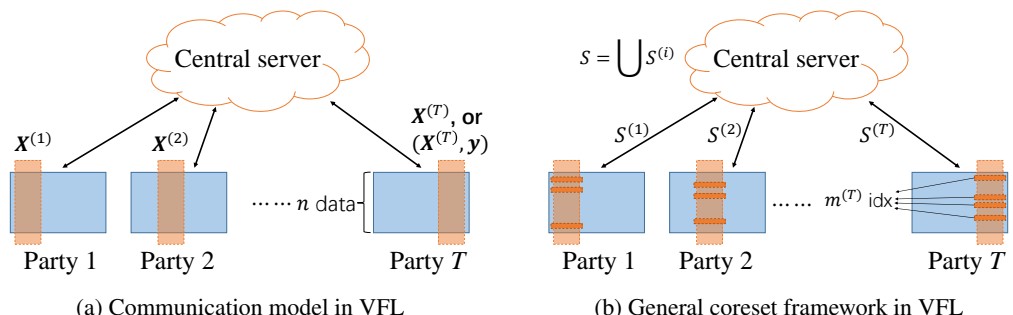

Figure 1: Illustration of coreset construction in VFL

property of coresets, there have been studies on coreset construction in the distributed/horizontal setting [2, 58, 1, 51]. To our knowledge, we are the first to consider coreset construction in VFL.

## 2 Problem Formulation/Model

In this section, we formally define our problems: coresets for vertical regularized linear regression and coresets for vertical $k$-means clustering (Problem 1).

**Vertical federated learning model.** We first introduce the model of vertical federated learning (VFL). Let $\boldsymbol{X} \subset \mathbb{R}^d$ be a dataset of size $n$ that is vertically separated stored in $T$ data parties ($T \geq 2$). Concretely, we represent each point $\boldsymbol{x}_i \in \boldsymbol{X}$ by $\boldsymbol{x}_i = (\boldsymbol{x}_i^{(1)}, \ldots, \boldsymbol{x}_i^{(T)})$ where $\boldsymbol{x}_i^{(j)} \in \mathbb{R}^{d_j}$ ($j \in [T]$), and each party $j \in [T]$ holds a local dataset $\boldsymbol{X}^{(j)} = \left\{ \boldsymbol{x}_i^{(j)} \right\}_{i \in [n]}$. Note that $\sum_{j \in [T]} d_j = d$. Additionally, if there is a label $y_i \in \mathbb{R}$ for each point $\boldsymbol{x}_i \in \boldsymbol{X}$, we assume the label vector $\boldsymbol{y} \in \mathbb{R}^n$ is stored in Party $T$. The objective of vertical federated learning is to collaboratively solve certain training problems in the central server with a total communication complexity as small as possible.

Similar to Ding et al. [19, Figure 1], we only allow the communication between the central server and each of the $T$ parties, and require the central server to hold the final solution. Note that the central server can also be replaced with any party in practice. For the communication complexity, we assume that transporting an integer/floating-point costs 1 unit, and consequently, transporting a $d$-dimensional vector costs $d$ communication units. See Figure 1a for an illustration.

**Vertical regularized linear regression and vertical $k$-means clustering.** In this paper, we consider the following two important machine learning problems in the VFL model.

**Definition 2.1** (**Vertical regularized linear regression (VRLR)**). Given a dataset $\boldsymbol{X} \subset \mathbb{R}^d$ together with labels $\boldsymbol{y} \in \mathbb{R}^n$ in the VFL model, a regularization function $R : \mathbb{R}^d \to \mathbb{R}_{\geq 0}$, the goal of the vertical regularized linear regression problem (VRLR) is to compute a vector $\boldsymbol{\theta} \in \mathbb{R}^d$ in the server that (approximately) minimizes $\text{cost}^R(\boldsymbol{X}, \boldsymbol{\theta}) := \sum_{i \in [n]} \text{cost}_i^R(\boldsymbol{X}, \boldsymbol{\theta}) = \sum_{i \in [n]} (\boldsymbol{x}_i^\top \boldsymbol{\theta} - \boldsymbol{y}_i)^2 + R(\boldsymbol{\theta})$, and the total communication complexity is as small as possible.

**Definition 2.2** (**Vertical $k$-means clustering (VKMC)**). Given a dataset $\boldsymbol{X} \subset \mathbb{R}^d$ in the VFL model, an integer $k \geq 1$, let $\mathcal{C}$ denote the collection of all $k$-center sets $\boldsymbol{C} \in \mathcal{C}$ with $|\boldsymbol{C}| = k$ and $d(\cdot, \cdot)$ denote the Euclidean distance. The goal of the vertical $k$-means clustering problem (VKMC) is to compute a $k$-center set $\boldsymbol{C} \in \mathcal{C}$ in the server that (approximately) minimizes $\text{cost}^C(\boldsymbol{X}, \boldsymbol{C}) := \sum_{i \in [n]} \text{cost}_i^C(\boldsymbol{X}, \boldsymbol{C}) = \sum_{i \in [n]} d(\boldsymbol{x}_i, \boldsymbol{C})^2 = \sum_{i \in [n]} \min_{\boldsymbol{c} \in \boldsymbol{C}} d(\boldsymbol{x}_i, \boldsymbol{c})^2$, and the total communication complexity is as small as possible.

Ding et al. [19] proposed a similar vertical $k$-means clustering problem and provided constant approximation schemes. They additionally compute an assignment from all points $x_i$ to solution $C$, which requires a communication complexity of at least $\Omega(nT)$. Due to huge $n$, directly solving VRLR or VKMC is a non-trivial task and may need a large communication complexity. To this end, we introduce a powerful data-reduction technique, called *coresets* [27, 22, 23].

**Coresets for VRLR and VKMC.** Roughly speaking, a coreset is a small summary of the original dataset, that approximates the learning objective for every possible choice of learning parameters. We first define coresets for offline regularized linear regression and $k$-means clustering as follows. As mentioned in Section 1.1, both problems have been well studied in the literature [22, 7, 12, 31, 15, 16].

**Definition 2.3** (**Coresets for offline regularized linear regression**). Given a dataset $\boldsymbol{X} \subset \mathbb{R}^d$ together with labels $\boldsymbol{y} \in \mathbb{R}^n$ and $\varepsilon \in (0,1)$, a subset $S \subseteq [n]$ together with a weight function $w : S \to \mathbb{R}_{\geq 0}$ is called an $\varepsilon$-coreset for offline regularized linear regression if for any $\boldsymbol{\theta} \in \mathbb{R}^d$,

$$\text{cost}^R(S, \boldsymbol{\theta}) := \sum_{i \in S} w(i) \cdot (\boldsymbol{x}_i^\top \boldsymbol{\theta} - y_i)^2 + R(\boldsymbol{\theta}) \in (1 \pm \varepsilon) \cdot \text{cost}^R(\boldsymbol{X}, \boldsymbol{\theta}).$$

**Definition 2.4** (**Coresets for offline $k$-means clustering**). Given a dataset $\boldsymbol{X} \subset \mathbb{R}^d$, an integer $k \geq 1$ and $\varepsilon \in (0,1)$, a subset $S \subseteq [n]$ together with a weight function $w : S \to \mathbb{R}_{\geq 0}$ is called an $\varepsilon$-coreset for offline $k$-means clustering if for any $k$-center set $\boldsymbol{C} \subset \mathbb{R}^d$,

$$\text{cost}^C(S, \boldsymbol{C}) := \sum_{i \in S} w(i) \cdot d(\boldsymbol{x}_i, \boldsymbol{C})^2 \in (1 \pm \varepsilon) \cdot \text{cost}^C(\boldsymbol{X}, \boldsymbol{C}).$$

Now we are ready to give the following main problem.

**Problem 1** (**Coreset construction for VRLR and VKMC**). Given a dataset $\boldsymbol{X} \subset \mathbb{R}^d$ (together with labels $\boldsymbol{y} \in \mathbb{R}^n$) in the VFL model and $\varepsilon \in (0,1)$, our goal is to construct an $\varepsilon$-coreset for regularized linear regression (or $k$-means clustering) in the server, with as small communication complexity as possible. See Figure 1b for an illustration.

Note that our coreset is a subset of indices which is slightly different from that in previous work [27, 22, 23], whose coreset consists of weighted points. This is because we would like to reduce data transportation from parties to the server due to privacy considerations. Specifically, if the communication schemes for VRLR and VKMC do not need to make data transportation, then we can avoid data transportation by first applying our coreset construction scheme and then doing the communication schemes based on the coreset. Moreover, we have the following theorem that shows how coresets reduce the communication complexity in the VFL models, and the proof is in Section C.

**Theorem 2.5** (**Coresets reduce the communication complexity for VRLR and VKMC**). *Given* $\varepsilon \in (0,1)$*, suppose there exist*

1. *a communication scheme $A$ that given a (weighted) dataset $\boldsymbol{X} \subset \mathbb{R}^d$ together with labels $\boldsymbol{y} \in \mathbb{R}^n$ in the VFL model, computes an $\alpha$-approximate solution ($\alpha \geq 1$) for VRLR (or VKMC) in the server with a communication complexity $\Lambda(n)$;*

2. *a communication scheme $A'$ that given a (weighted) dataset $\boldsymbol{X} \subset \mathbb{R}^d$ together with labels $\boldsymbol{y} \in \mathbb{R}^n$ in the VFL model, constructs an $\varepsilon$-coreset for VRLR (or VKMC respecitively) of size $m$ in the server with a communication complexity $\Lambda_0$.*

*Then there exists a communication scheme that given a (weighted) dataset $\boldsymbol{X} \subset \mathbb{R}^d$ together with labels $\boldsymbol{y} \in \mathbb{R}^n$ in the VFL model, computes an $(1 + 3\varepsilon)\alpha$-approximate solution ($\alpha \geq 1$) for VRLR (or VKMC respectively) in the server with a communication complexity $\Lambda_0 + 2mT + \Lambda(m)$.*

Usually, $\Lambda(m) = \Omega(mT)$ and $\Lambda_0$ is small or comparable to $Tm$ (see Theorems 4.2 and 5.2 for examples). Consequently, the total communication complexity by introducing coresets is dominated by $\Lambda(m)$, which is much smaller compared to $\Lambda(n)$. Hence, coreset can efficiently reduce the communication complexity with a slight sacrifice on the approximate ratio.

## 3 A Unified Scheme for VFL Coresets via Importance Sampling

In this section, we propose a unified communication scheme (Algorithm 1) that will be used as a meta-algorithm for solving Problem 1. We assume each party $j \in [T]$ holds a real number $g_i^{(j)} \geq 0$ for data $\boldsymbol{x}_i^{(j)}$ in Algorithm 1, that will be computed locally for both VRLR (Algorithm 2) and VKMC (Algorithm 3). There are three communication rounds in Algorithm 1. In the first round (Lines 2-4), the server knows all "local total sensitivities" $\mathcal{G}^{(j)}$, takes samples of $[T]$ with probability proportional

to $\mathcal{G}^{(j)}$, and sends $a_j$ to each party $j$, where $a_j$ is the number of local samples of party $j$ for the second round. In the second round (Lines 5-6), each party samples a collection $S^{(j)} \subseteq [n]$ of size $a_j$ with probability proportional to $g_i^{(j)}$. The server achieves the union $S = \bigcup_{j \in [T]} S^{(j)}$. In the third round (Lines 7-8), the goal is to compute weights $w(i)$ for all samples. In the end, we achieve a weighted subset $(S, w)$. We propose the following theorem to analyze the performance of Algorithm 1 and show that $(S, w)$ is a coreset when size $m$ is large enough.

**Theorem 3.1** (**The performance of Algorithm 1**). *The communication complexity of Algorithm 1 is* $O(mT)$. *Let* $\varepsilon, \delta \in (0, 1/2)$ *and* $k \geq 1$ *be an integer. We have*

- *Let* $\zeta = \max_{i \in [n]} \sup_{\boldsymbol{\theta} \in \mathbb{R}^d} \frac{\mathrm{cost}_i^R(\boldsymbol{X}, \boldsymbol{\theta})}{\mathrm{cost}^R(\boldsymbol{X}, \boldsymbol{\theta})} / \sum_{j \in [T]} g_i^{(j)}$ *and* $m = O\left(\varepsilon^{-2} \zeta \mathcal{G}(d^2 \log(\zeta\mathcal{G}) + \log(1/\delta))\right)$. *With probability at least* $1 - \delta$, $(S, w)$ *is an* $\varepsilon$*-coreset for offline regularized linear regression.*

- *Let* $\zeta = \max_{i \in [n]} \sup_{\boldsymbol{C} \in \mathcal{C}} \frac{\mathrm{cost}_i^C(\boldsymbol{X}, \boldsymbol{C})}{\mathrm{cost}^C(\boldsymbol{X}, \boldsymbol{C})} / \sum_{j \in [T]} g_i^{(j)}$ *and* $m = O\left(\varepsilon^{-2} \zeta \mathcal{G}(dk \log(\zeta\mathcal{G}) + \log(1/\delta))\right)$. *With probability at least* $1 - \delta$, $(S, w)$ *is an* $\varepsilon$*-coreset for offline* $k$*-means clustering.*

The proof can be found in Appendix D. The main idea is to show that Algorithm 1 simulates a well-known importance sampling framework for offline coreset construction by [22, 7]. The term $\sup_{\boldsymbol{\theta} \in \mathbb{R}^d} \frac{\mathrm{cost}_i^R(\boldsymbol{X}, \boldsymbol{\theta})}{\mathrm{cost}^R(\boldsymbol{X}, \boldsymbol{\theta})}$ (or $\sup_{\boldsymbol{C} \in \mathcal{C}} \frac{\mathrm{cost}_i^C(\boldsymbol{X}, \boldsymbol{C})}{\mathrm{cost}^C(\boldsymbol{X}, \boldsymbol{C})}$) is called the *sensitivity* of point $\boldsymbol{x}_i$ for VRLR (or VKMC) that represents the maximum contribution of $\boldsymbol{x}_i$ over all possible parameters. Algorithm 1 aims to use $\sum_{j \in [T]} g_i^{(j)}$ to estimate the sensitivity of $\boldsymbol{x}_i$, and hence, $\zeta$ represents the maximum sensitivity gap over all points. The performance of Algorithm 1 mainly depends on the quality of these estimations $\sum_{j \in [T]} g_i^{(j)}$. As both $\zeta$ and the total sum $\mathcal{G} = \sum_{i \in [n], j \in [T]} g_i^{(j)}$ become smaller, the required size $m$ becomes smaller. Specifically, if both $\zeta$ and $\mathcal{G}$ only depends on parameters $k, d, T$, the coreset size $m$ is independent of $n$ as expected. Combining with Theorem 2.5, we can heavily reduce the communication complexity for VRLR or VKMC.

---

**Algorithm 1** A unified importance sampling for coreset construction in the VFL model

---

**Input:** Each party $j \in [T]$ holds data $\boldsymbol{x}_i^{(j)}$ together with a real number $g_i^{(j)} \geq 0$, an integer $m \geq 1$
**Output:** a weighted collection $S \subseteq [n]$ of size $|S| \leq m$

1: **procedure** DIS$(m, \{g_i^{(j)} : i \in [n], j \in [T]\})$
2:     Each party $j \in [T]$ sends $\mathcal{G}^{(j)} \leftarrow \sum_{i \in [n]} g_i^{(j)}$ to the server.         ▷ 1st round begins
3:     The server computes $\mathcal{G} = \sum_{j \in [T]} \mathcal{G}^{(j)}$ and samples a multiset $A \subseteq [T]$ of $m$ samples, where each sample $j \in [T]$ is selected with probability $\mathcal{G}^{(j)}/\mathcal{G}$.
4:     The server sends $a_j \leftarrow \#j \in A$ to each party $j \in [T]$.         ▷ 1st round ends
5:     Each party $j \in [T]$ samples a multiset $S^{(j)} \subseteq [n]$ of size $a_j$, where each sample $i \in [n]$ is selected with probability $g_i^{(j)}/\mathcal{G}^{(j)}$, and sends $S^{(j)}$ to the server.         ▷ 2nd round begins
6:     The server broadcasts a multiset $S \leftarrow \bigcup_{j \in [T]} S^{(j)}$ to all parties.         ▷ 2nd round ends
7:     Each party $j \in [T]$ sends $G^{(j)} = \left\{ g_i^{(j)} : i \in S \right\}$ to the server.         ▷ 3rd round begins
8:     The server computes weights $w(i) \leftarrow \mathcal{G}/|S| \cdot \sum_{j \in [T]} g_i^{(j)}$ for each $i \in S$.         ▷ 3rd round ends
9:     **return** $(S, w)$
10: **end procedure**

---

**Privacy issue.** We consider the privacy of the proposed scheme from two aspects: coreset construction and model training. As for the coreset construction part (Algorithm 1), the privacy leakage comes from the "sensitivity score" $g_i^{(j)}$ of the data points in different parties. To tackle this problem, we can use secure aggregation [5] to transport the sum $g_i = \sum_{j=1}^{T} g_i^{(j)}$ to the server without revealing the exact values of $g_i^{(j)}$s (Line 7 of Algorithm 1). The server only knows $(S, w)$ and $\mathcal{G}^{(j)}$s. For the model training part, we can apply the secure VFL algorithms if existed, e.g., using homomorphic encryption on SAGA for regression (it is an extension from SGD to SAGA [28]).

---

**Algorithm 2** Vertical federated coreset construction for Regularized Linear Regression (VRLR)

---

**Input:** Each party $j \in [T]$ holds the data $\boldsymbol{x}_i^{(j)}$ for all $i \in [n]$, coreset size $m$.

 1: **for** each party $j \in [T]$ **do**

 2:     Compute orthornormal basis $\boldsymbol{U}^{(j)} = [\boldsymbol{u}_1^{(j)}, \dots, \boldsymbol{u}_n^{(j)}]^\top$ of $\boldsymbol{X}^{(j)}$

 3:     $g_i^{(j)} \leftarrow \|\boldsymbol{u}_i^{(j)}\|^2 + \frac{1}{n}$ for all $i \in [n]$

 4: **end for**

 5: **return** $(S, w) \leftarrow \mathtt{DIS}(m, \{g_i^{(j)}\})$

---

Note that the VFL communication model in Section 2 is assumed to be semi-honest. Suppose some party $j$ is malicious, then it can report a large enough $\mathcal{G}^{(j)}$ (Line 2 of Algorithm 1) such that the server sets the number of samples $a_j \approx m$ in party $j$ (Line 4 of Algorithm 1). Consequently, party $j$ can sample a large multi-set $S^{(j)}$ which heavily affects the resulting set $S$. For instance, by reporting $S^{(j)}$ of uniform samples, party $j$ can make $S$ close to uniform sampling and loss the theoretical guarantees in Theorem 3.1.

## 4   Coreset Construction for VRLR

In this section, we discuss the coreset construction for VRLR. We first show that it is generally hard to construct a strong coreset for VRLR. Then, we show how to communication-efficiently construct coresets for VRLR under mild assumption. All missing proofs can be found in Section E.

With slightly abuse of notation, we denote $\boldsymbol{X} \in \mathbb{R}^{n \times d}$, $\boldsymbol{X}^{(j)} \in \mathbb{R}^{n \times d_j}$ to be the data matrix of whole data and the data matrix stored on party $j$ respectively. Since there are labels $\boldsymbol{y}$ stored on party $T$, $\boldsymbol{X}^{(T)}$ has dimension $n \times (d_T + 1)$.

**Communication complexity lower bound for VRLR**   We first show that it is *hard* to compute the coreset for VRLR by proving an $\Omega(n)$ deterministic communication complexity lower bound.

**Theorem 4.1** (**Communication complexity of coreset construction for VRLR**). *Let $T \geq 2$. Given constant $\varepsilon \in (0, 1)$, any deterministic communication scheme that constructs an $\varepsilon$-coreset for VRLR requires a communication complexity $\Omega(n)$.*

The communication complexity lower bound for linear regression has also been considered in the HFL setting [67], e.g., Vempala et al. [67] also gets a deterministic communication complexity lower bound. Theorem 4.1 shows that linear regression in the VFL setting is "hard" and thus we may need to add data assumptions to get theoretical guarantees for coreset construction.

**Communication-efficient coreset construction for VRLR**   Now we show that under mild condition, we can construct a strong coreset for VRLR using $o(n)$ number of communication. Specifically, we assume the data $\boldsymbol{X}$ satisfies the following assumption, which will be justified in the appendix.

**Assumption 4.1.** *Let $\boldsymbol{U}^{(j)} \in \mathbb{R}^{n \times d_j'}$ denote the orthonormal basis of the column space of $\boldsymbol{X}^{(j)}$ stored on party $j$ ($\boldsymbol{U}^{(T)}$ denotes the orthonormal basis of $[\boldsymbol{X}^{(T)}, y]$), and then the matrix $\boldsymbol{U} = [\boldsymbol{U}^{(1)}, \boldsymbol{U}^{(2)}, \dots, \boldsymbol{U}^{(T)}]$ has smallest singular value $\sigma_{\min}(\boldsymbol{U}) \geq \gamma > 0$.*

Intuitively, $\gamma \in (0, 1]$ represents the degree of orthonormal among data in different parties. As the larger $\gamma$ is, the more orthonormal among the column spaces of $X^{(j)}$, and thus $U$ is more close to the orthonormal basis computed on $X$ directly. Now we introduce our coreset construction algorithm for VRLR (Algorithm 2). At a very high level perspective, we let each party $j$ to compute a coreset $S^{(j)}$ based on its own data $\boldsymbol{X}^{(j)}$, and combine all the $S^{(j)}$ together to obtain a final coreset $S$. More specifically, for each party $j$, we let it to compute $\boldsymbol{U}^{(j)} = [\boldsymbol{u}_1^{(j)}, \dots, \boldsymbol{u}_n^{(j)}]^\top$ based on the data $\boldsymbol{X}^{(j)}$, and set $g_i^{(j)} = \|\boldsymbol{u}_i^{(j)}\|^2 + \frac{1}{n}$ to be the weight of data $i$ on party $j$. Then, we set $g_i = \sum_{j \in [T]} g_i^{(j)}$ to be the final weight of data $\boldsymbol{x}_i$ and want to sample $m$ samples using weight $g_i$. To do this, we apply the DIS procedure (Algorithm 1).

**Theorem 4.2** (**Coresets for VRLR**). *For a given dataset $\boldsymbol{X} \subset \mathbb{R}^d$ satisfying Assumption 4.1, number of parties $T \geq 1$ and constants $\varepsilon, \delta \in (0, 1)$, with probability at least $1 - \delta$, Algorithm 2 constructs*

*an $\varepsilon$-coreset for VRLR of size $m = O(\varepsilon^{-2}\gamma^{-2}d(d^2 \log \gamma^{-2}d + \log 1/\delta))$, and uses communication complexity $O(mT)$.*

Note that the coreset size and the total communication are all independent on $n$, and thus when combined with Theorem 2.5, using coreset construction can reduce the communication complexity for VRLR. When Assumption 4.1 is not satisfied, Algorithm 4.2 is not guaranteed to return a strong coreset. However, as shown in the following remark, it will return another kind of coreset called *robust coreset* [22, 32, 69], which allows a small portion of data to be treated as outliers and excluded both in $S$ and $\boldsymbol{X}$ when evaluating the quality of $S$. The outliers represent a small percentage of data with unbounded sensitivity gap. More details can be found in the Theorem G.3.

**Remark 4.3** (Robust coreset for VRLR). *Given a dataset $\boldsymbol{X} \subset \mathbb{R}^d$ together with labels $\boldsymbol{y} \in \mathbb{R}^n$, $\varepsilon \in (0,1)$ and $\beta \in [0,1)$, a subset $S \subseteq [n]$ together with a weight function $w : S \to \mathbb{R}_{\geq 0}$ is called a $(\beta, \varepsilon)$-robust coreset for offline regularized linear regression if for any $\boldsymbol{\theta} \in \mathbb{R}^d$, there exists a subset $O_{\boldsymbol{\theta}} \subseteq [n]$ such that $|O_{\boldsymbol{\theta}}|/n \leq \beta$, $|S \cap O_{\boldsymbol{\theta}}|/|S| \leq \beta$ and*

$$\mathsf{cost}^R(S \backslash O_{\boldsymbol{\theta}}, \boldsymbol{\theta}) \in \mathsf{cost}^R(\boldsymbol{X} \backslash O_{\boldsymbol{\theta}}, \boldsymbol{\theta}) \pm \varepsilon \cdot \mathsf{cost}^R(\boldsymbol{X}, \boldsymbol{\theta}).$$

*If Assumption 4.1 is not satisfied, for $m = O((\varepsilon\beta T)^{-2}d^6)$, Algorithm 2 will return a $(\beta, \varepsilon)$-robust coreset for VRLR with communication complexity $O(mT)$.*

# 5   Coreset Construction for VKMC

In this section, we discuss the coreset construction for VKMC. Similar to VRLR, we first show it generally requires $\Omega(n)$ communication complexity to construct a coreset for VKMC, and then we show that it is possible to vastly reduce the communication complexity (Algorithm 3) under mild data assumption. All missing proofs can be found in Section F.

**Communication complexity lower bound for VKMC.**   We first present an $\Omega(n)$ communication complexity lower bound for constructing an $\varepsilon$-coreset for VKMC in the following theorem.

**Theorem 5.1** (**Communication complexity of coreset construction for VKMC**). *Let $d \geq T \geq 2$. Given a constant $\varepsilon \in (0,1)$ and an integer $k \geq 3$, any randomized communication scheme that constructs an $\varepsilon$-coreset for VKMC with probability 0.99 requires a communication complexity $\Omega(n)$.*

Different from VRLR, we have a randomized communication complexity lower bound for VKMC. Similarly, we also need to introduce certain data assumptions to get theoretical guarantees for coreset construction due to this hardness result.

**Communication-efficient coreset construction for VKMC**   Now we show how to communication-efficiently construct coresets for VKMC under mild condition. Specifically, we assume that the data satisfies the following assumption, which will be justified in the appendix.

**Assumption 5.1.** *There exists $\tau \geq 1$ and some party $t \in [T]$ such that $\|\boldsymbol{x}_i - \boldsymbol{x}_j\|^2 \leq \tau \left\|\boldsymbol{x}_i^{(t)} - \boldsymbol{x}_j^{(t)}\right\|^2$ for any $i, j \in [n]$.*

This assumption says that, there is a party that is "important", and any two data points which can be differentiated can also be differentiated on that party to some extent. Specifically, as $\tau$ is more close to 1, Assumption 5.1 implies that there exists a party $t \in [T]$ whose local pairwise distances $\|x_i^{(t)} - x_j^{(t)}\|$s are close to the corresponding global pairwise distances $\|x_i - x_j\|$s. Then we introduce our coreset construction algorithm for VKMC (Algorithm 3). For the input, note that there exist several constant approximation algorithms for $k$-means [41, 66]. The widely used $k$-means++ algorithm [66] provides an $O(\ln k)$-approximation and performs well in practice. Similar to Algorithm 2 for VRLR, Algorithm 3 also applies Algorithm 1 after computing $g_i^{(j)}$ locally. The key is to construct local sensitivities $g_i^{(j)}$ to upper bound both $\zeta$ and $\mathcal{G}$ in Theorem 3.1. The derivation of the local sensitivities $g_i^{(j)}$ defined in Line 10 is partly inspired by [65], which upper bounds the total sensitivity of a point set in clustering problems by projecting points onto an optimal solution. Intuitively, if some party $t$ satisfies Assumption 5.1, a constant factor approximate solution computed locally in party $t$ can also induce a global one. Then by projecting points onto this global constant

**Algorithm 3** Vertical federated coreset construction for $k$-means Clustering (VKMC)

---

**Input:** Each party $j \in [T]$ holds the data $\boldsymbol{x}_i^{(j)}$ for all $i \in [n]$, coreset size $m$, number of centers $k$, an $\alpha$-approximation algorithm $\mathcal{A}$ (e.g. $k$-means++).

**Output:** a weighted collection $S \subseteq [n]$ of size $|S| \leq m$

1: **for all** party $j \in [T]$ **do**
2:     $\boldsymbol{C}^{(j)} \leftarrow \mathcal{A}(\{\boldsymbol{x}_i^{(j)}\}_{i \in [n]})$. Note that $\boldsymbol{C}^{(j)} = \{\boldsymbol{c}_1^{(j)}, \boldsymbol{c}_2^{(j)}, \ldots, \boldsymbol{c}_k^{(j)}\}$.
3:     Initialize $\boldsymbol{B}_l^{(j)} = \varnothing$ for $l \in [k]$.
4:     **for all** $i \in [n]$ **do**
5:         $\pi(i) \leftarrow \arg \min_{l \in [k]} d(\boldsymbol{x}_i^{(j)}, \boldsymbol{c}_l^{(j)})$          ▷ a mapping to find the closest center locally.
6:         $\boldsymbol{B}_{\pi(i)}^{(j)} \leftarrow \boldsymbol{B}_{\pi(i)}^{(j)} \cup i$.
7:     **end for**
8:     $\mathrm{cost}^{(j)} \leftarrow \sum_{i \in [n]} d(\boldsymbol{x}_i^{(j)}, \boldsymbol{C}^{(j)})^2$          ▷ $d(\boldsymbol{x}_i^{(j)}, \boldsymbol{C}^{(j)}) = d(\boldsymbol{x}_i^{(j)}, \boldsymbol{c}_{\pi(i)}^{(j)})$
9:     **for all** $i \in [n]$ **do**
10:         $l \leftarrow \pi(i)$, $g_i^{(j)} \leftarrow \frac{\alpha d(\boldsymbol{x}_i^{(j)}, \boldsymbol{c}_l^{(j)})^2}{\mathrm{cost}^{(j)}} + \frac{\alpha \sum_{i' \in \boldsymbol{B}_l^{(j)}} d(\boldsymbol{x}_{i'}^{(j)}, \boldsymbol{c}_l^{(j)})^2}{|\boldsymbol{B}_l^{(j)}| \mathrm{cost}^{(j)}} + \frac{2\alpha}{|\boldsymbol{B}_l^{(j)}|}$.
11:     **end for**
12: **end for**
13: **return** $(S, w) \leftarrow \mathtt{DIS}(m, \{g_i^{(j)}\})$

---

approximation, we can prove that $g_i^{(t)}$ (scaled by some constant factor) is an upper bound of the global sensitivity of $\boldsymbol{x}_i$ for any $i \in [n]$. Though unaware of which party satisfies Assumption 5.1, it suffices to sum up $g_i^{(j)}$ over $j \in [T]$, only costing an additional $T$ in $\mathcal{G}$. Finally, we can upper bound $\zeta$ by $O(\tau)$ and $\mathcal{G}$ by $O(kT)$ respectively. The main theorem is as follows.

**Theorem 5.2** (Coresets for VKMC). *For a given dataset $\boldsymbol{X} \subset \mathbb{R}^d$ satisfying Assumption 5.1, an $\alpha$-approximation algorithm for $k$-means with $\alpha = O(1)$, integers $k \geq 1$, $T \geq 1$ and constants $\varepsilon, \delta \in (0, 1)$, with probability at least $1 - \delta$, Algorithm 3 constructs an $\varepsilon$-coreset for VKMC of size $m = O(\varepsilon^{-2} \alpha \tau k T (dk \log (\alpha \tau k T) + \log 1/\delta))$, and uses communication complexity $O(mT)$.*

Again, note that both the coreset size and communication complexity are independent of $n$. Thus, using Algorithm 3 together with other baseline algorithms can drastically reduce the communication complexity. Similar to VRLR, we have the following remark when the data assumption (Assumption 5.1) is not satisfied. More details can be found in the Theorem G.4.

**Remark 5.3** (Robust coreset for VKMC). *Given a dataset $\boldsymbol{X} \subset \mathbb{R}^d$, an integer $k \geq 1$, $\varepsilon \in (0, 1)$ and $\beta \in [0, 1)$, a subset $S \subseteq [n]$ together with a weight function $w : S \to \mathbb{R}_{\geq 0}$ is called a $(\beta, \varepsilon)$-robust coreset for offline $k$-means clustering if for any $\boldsymbol{C} \subset \mathbb{R}^d$, there exists a subset $O_{\boldsymbol{C}} \subseteq [n]$ such that $|O_{\boldsymbol{C}}|/n \leq \beta$, $|S \cap O_{\boldsymbol{C}}|/|S| \leq \beta$ and*

$$\mathrm{cost}^C(S \backslash O_{\boldsymbol{C}}, \boldsymbol{C}) \in \mathrm{cost}^C(\boldsymbol{X} \backslash O_{\boldsymbol{C}}, \boldsymbol{C}) \pm \varepsilon \cdot \mathrm{cost}^C(\boldsymbol{X}, \boldsymbol{C}).$$

*If Assumption 5.1 is not satisfied, for $m = O((\varepsilon\beta)^{-2} k^5 d)$ Algorithm 3 will return a $(\beta, \varepsilon)$-robust coreset for VKMC with communication complexity $O(mT)$.*

## 6 Numerical Experiments

In this section, we present the numerical experiments, which corroborate our theoretical results. We conduct experiments on a single system that simulates the distributed settings.[3]

**Empirical setup.** We conduct experiments on the `YearPredictionMSD` dataset [4] for both VRLR and VKMC. `YearPredictionMSD` dataset has 515345 data, and each data contains 90 features and a corresponding label. We assume there are $T = 3$ parties and each party stories 30 distinct features. For VRLR, we split the data into a training set with size 463715 and a testing set with size 51630. We consider ridge regression in VRLR by letting $R(\boldsymbol{\theta}) = \lambda \|\boldsymbol{\theta}\|^2$ for $\lambda = 0.1n$ where $n$ is the dataset

---

[3]The codes are available at `https://github.com/haoyuzhao123/coreset-vfl-codes`.

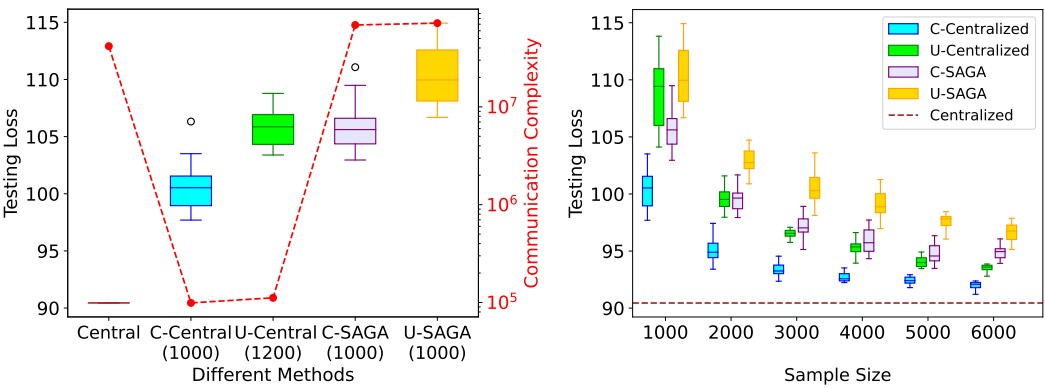

Figure 2: Left: Testing loss and communication complexity of VRLR for different methods. C and U means using coreset or uniform sampling. The number in the parentheses denotes the sample size. Right: Testing loss of VRLR for different methods under multiple sample sizes.

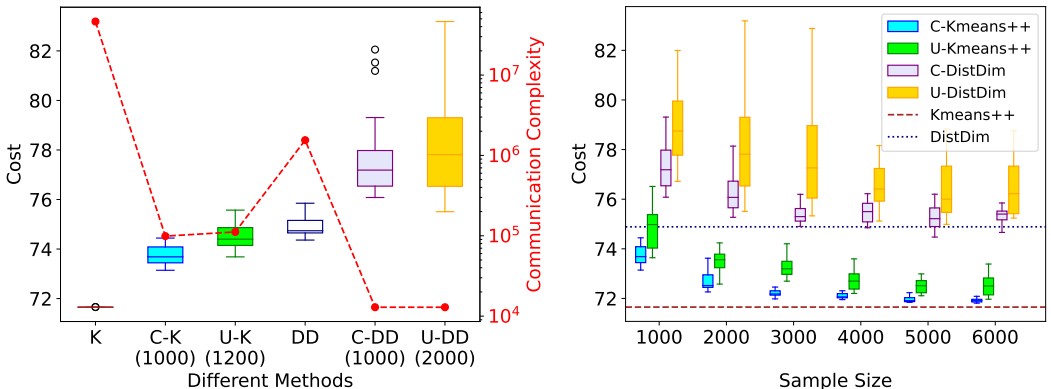

Figure 3: Left: Cost and communication complexity of VKMC for different methods. C and U means using coreset sampling or uniform sampling. The number in the parentheses denote the sample size. Right: Cost of VKMC for different methods under multiple sample sizes.

size. For VKMC, there is only one training set with size 515345 and without labels. We choose $k = 10$ (10 centers) and we normalize each feature with mean 0 and standard deviation 1 for VKMC.

For VRLR, we consider two baselines: 1) CENTRAL as the procedure that transfers all data to the central server and solves the problem using scikit-learn package [57]; 2) SAGA as using [18]'s algorithm to optimize in a VFL fashion. For VKMC, we also consider two baselines: 1) KMEANS++ as the procedure that transfers all data to the central server and clusters using KMEANS++ [66]; 2) DISTDIM by [19].

For each baseline, we compare our coreset algorithm with uniform sampling. We use C-X to denote coreset sampling followed by algorithm X and U-X for uniform sampling followed by algorithm X, e.g. C-DISTDIM means that we apply coreset construction and then use DISTDIM algorithm. We compare C-X and U-X with different sizes, and each experiment is repeated 20 times.

**Empirical results.** Figure 2 shows our results for VRLR and Figure 3 shows our results for VKMC. Table 1 summarize the results. For VRLR, since it is a supervised learning problem, we report the testing loss; for VKMC, it is an unsupervised learning task and the cost refers to the training loss on the full training data.

**Coreset sampling performs close to the baseline with less communication.** From the results, we find that using our coreset can achieve a similar loss compared to the baseline, while the communication complexity is reduced drastically. Specifically by Table 1, our coreset algorithm C-CENTRAL can

Table 1: Results of VRLR and VKMC on `YearPredictionMSD` dataset. Left: results for VRLR. Right: results for VKMC. The average and std. are computed using the 20 repeated experiments. The communication complexity denotes the average communication complexity, and the number in the parenthesis denotes the fraction of coreset construction (or uniform sampling respectively).

| Alg (size) | Cost avg/std | Com. compl. | Alg | Cost avg/std | Com. compl. | Alg (size) | Cost avg/std | Com. compl. | Alg | Cost avg/std | Com. compl. |
|---|---|---|---|---|---|---|---|---|---|---|---|
| CENTRAL | 90.45/0.00 | 4.2e7 | | | | KMEANS++ | 71.65/0.00 | 4.6e7 | | | |
| 1000 | 100.50/2.11 | 9.9e4(0.09) | | 108.79/2.79 | 9.3e4(0.03) | 1000 | 73.76/0.38 | 9.9e4(0.09) | | 74.91/0.81 | 9.3e4(0.03) |
| 2000 | 95.01/0.95 | 2.0e5(0.09) | | 99.65/1.01 | 1.9e5(0.03) | 2000 | 72.68/0.35 | 2.0e5(0.09) | | 73.52/0.44 | 1.9e5(0.03) |
| 3000 | 93.39/0.63 | 3.0e5(0.09) | | 96.68/0.73 | 2.8e5(0.03) | 3000 | 72.23/0.17 | 3.0e5(0.09) | | 73.25/0.39 | 2.8e5(0.03) |
| 4000 | 92.73/0.41 | 4.0e5(0.09) | C-CENTRAL / U-CENTRAL | 95.32/0.65 | 3.7e5(0.03) | 4000 | 72.14/0.19 | 4.0e5(0.09) | C-KMEANS++ / U-KMEANS++ | 72.74/0.39 | 3.7e5(0.03) |
| 5000 | 92.42/0.33 | 5.0e5(0.09) | | 94.08/0.48 | 4.7e5(0.03) | 5000 | 71.97/0.13 | 5.0e5(0.09) | | 72.54/0.37 | 4.7e5(0.03) |
| 6000 | 91.97/0.32 | 5.9e5(0.09) | | 93.54/0.44 | 5.6e5(0.03) | 6000 | 71.92/0.09 | 5.9e5(0.09) | | 72.57/0.44 | 5.6e5(0.03) |
| SAGA | N/A | N/A | | | | DISTDIM | 74.89/0.00 | 1.5e6 | | | |
| 1000 | 105.93/2.17 | 6.9e7(<0.01) | | 110.43/2.43 | 7.2e7(<0.01) | 1000 | 77.75/1.78 | 1.3e4(0.70) | | 78.87/1.44 | 7.0e3(0.43) |
| 2000 | 99.55/0.96 | 1.4e8(<0.01) | | 102.96/1.09 | 1.6e8(<0.01) | 2000 | 76.82/0.85 | 2.5e4(0.72) | | 78.13/2.10 | 1.3e4(0.47) |
| 3000 | 97.13/0.98 | 1.9e8(<0.01) | | 100.64/1.44 | 2.4e8(<0.01) | 3000 | 75.52/0.64 | 3.7e4(0.73) | | 77.85/2.23 | 1.9e4(0.48) |
| 4000 | 95.90/1.04 | 2.5e8(<0.01) | C-SAGA / U-SAGA | 99.10/1.20 | 3.3e8(<0.01) | 4000 | 75.49/0.45 | 4.9e4(0.74) | C-DISTDIM / U-DISTDIM | 76.70/1.27 | 2.5e4(0.48) |
| 5000 | 94.75/0.85 | 3.0e8(<0.01) | | 97.64/0.59 | 3.9e8(<0.01) | 5000 | 75.27/0.50 | 6.1e4(0.74) | | 76.38/1.17 | 3.1e4(0.49) |
| 6000 | 94.83/0.65 | 3.6e8(<0.01) | | 96.88/1.12 | 4.6e8(<0.01) | 6000 | 75.32/0.33 | 7.3e4(0.74) | | 76.44/1.03 | 3.7e4(0.49) |

use less than 0.4% of training data (2000/463715) and achieve a $95.01/90.45 \approx 1.05$-approximate solution for VRLR compared to the baseline CENTRAL. Observe that a larger coreset size leads to a smaller cost and a larger communication complexity. From Figures 2 and 3 (left), using coresets can reduce 50-100x communication complexity compared with the original baselines.

**Coreset performs better than uniform sampling under the same communication.** From Figures 2 and 3 (right), we observe that our coresets always achieve a better solution than uniform sampling under the same sample size. Table 1 also reflects this trend. Under the same sample size, the communication complexity by uniform sampling is slightly lower than that of coreset, since there is no need to transfer weights in uniform sampling. Thus, we also compare the performance of our coresets and uniform sampling under the same communication complexity. From Figures 2 and 3 (left), we find that for different baselines, our coreset algorithms still achieve better testing loss/training cost while using fewer or the same communication, compared to uniform sampling.

**Coreset and uniform sampling may also make the problem feasible.** It is also interesting to observe that SAGA will not converge (or very slowly) on the original VRLR problem (Table 1), possibly because of the large dataset and the ill-conditioned optimization problem. However, by applying the coreset/uniform sampling, SAGA works for VRLR. This also indicates the effectiveness of our framework and the importance to reduce the dependency on $n$ (the dataset size).

## 7  Conclusion and Future Directions

In this paper, we first consider coreset construction in the vertical federated learning setting. We propose a unified coreset framework for communication-efficient VFL, and apply the framework to two important learning tasks: regularized linear regression and $k$-means clustering. We verify the efficiency of our coreset algorithms both theoretically and empirically, which can drastically alleviate the communication complexity while still maintaining the solution quality.

Our work initializes the topic of introducing coresets to VFL, which leaves several future directions. Firstly, our VFL coreset size is still larger than that of offline coresets for both VRLR and VKMC, even under certain data assumptions. One direction is to further improve the coreset size. Another interesting direction is to extend coreset construction to other learning tasks in the VFL setting, e.g., logistic regression or gradient boosting trees.

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
