# Appendix

# A  Additional Experiments

In this section, we present some additional experiments. This section is organized as follow: in Section A.1, we conduct experiments using different number of parties (as opposed to three parties in Section 6); in Section A.2, we test our methods using other regularizer for VRLR, e.g., Lasso; in Section A.3, we test our methods in VKMC with different number of centers; and finally in Section A.4, we conduct experiments on another dataset (KC House Dataset [35]).

## A.1  Different number of parties

In this section, we test our algorithms using different number of parties. We choose to use five parties ($T = 5$) in this section instead of three parties in Section 6.

**Empirical setup**  Most of the experimental setups are the same as those in Section 6, except that now we use 5 parties instead of 3 parties. There are 90 dimensions for a single data in YearPredictionMSD dataset, and we let each party hold 18 dimensions. Besides, changing the number of parties does not affect the performance of U-Central and U-SAGA (but the number of communication will change due to different number of parties), and we reuse the results from Section 6 and recalculate the number of communications.

**Empirical results**  Figure 4 and 5 summarize our results for VRLR and VKMC respectively. Note that all the observations in Section 6 hold for 5 parties.

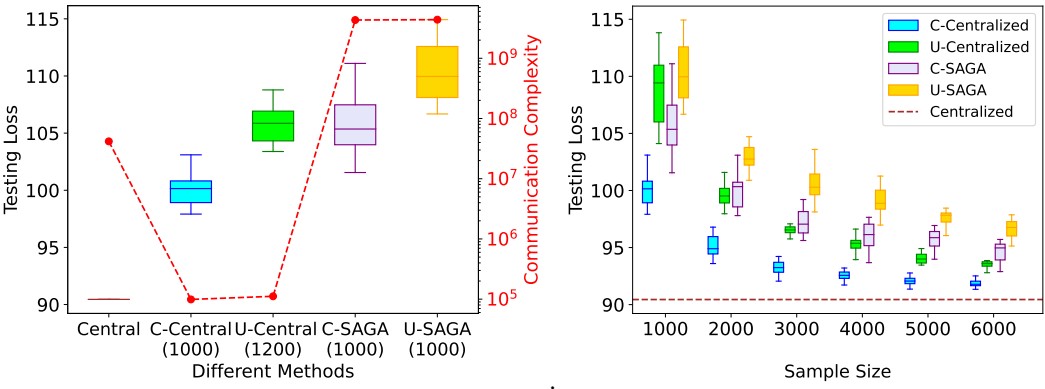

.

Figure 4: *Results for 5 parties (Section A.1)* Left: Testing loss and communication complexity of VRLR for different methods. C and U means using coreset or uniform sampling. The number in the parentheses denote the sample size. Right: Testing loss of VRLR for different methods under multiple sample sizes.

## A.2  Different regularizer for VRLR

In this part, we consider using different regularizers in VRLR.

**Empirical setup**  We consider three different regression problems: plain linear regression, Lasso regression, and elastic nets. In Section 6, we consider the Ridge regression ($R(\boldsymbol{\theta}) = 0.1n \left\| \boldsymbol{\theta} \right\|_2^2$ where $n$ is the dataset size), and in this part, linear regression denotes the optimization problem where $R(\boldsymbol{\theta}) = 0$, Lasso regression denotes the problem where $R(\boldsymbol{\theta}) = 2n \left\| \boldsymbol{\theta} \right\|_1$, and elastic net denotes the problem where $R(\boldsymbol{\theta}) = 2n \left\| \boldsymbol{\theta} \right\|_1 + n \left\| \boldsymbol{\theta} \right\|_2^2$. All the experiments setup remains the same as Section 6, except the for Lasso regression and elastic nets, there is no SAGA solver and we only compare C-Central and U-Central with Central.

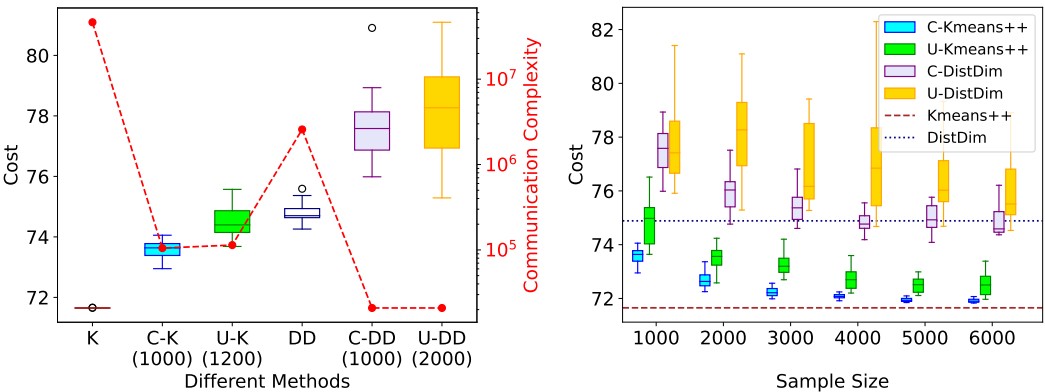

Figure 5: *Results for 5 parties (Section A.1)* Left: Cost and communication complexity of VKMC for different methods. C and U means using coreset sampling or uniform sampling. The number in the parentheses denote the sample size. Right: Cost of VKMC for different methods under multiple sample sizes.

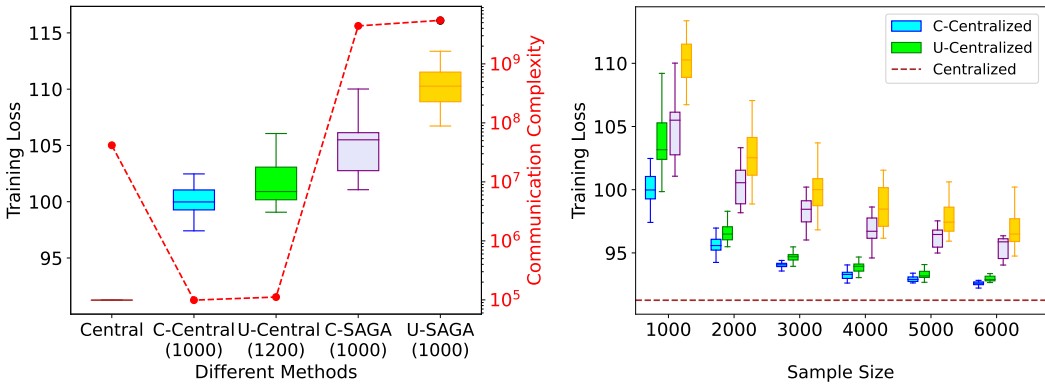

.

Figure 6: *Results for linear regression (Section A.2)* Left: Training loss and communication complexity of VRLR for different methods. C and U means using coreset or uniform sampling. The number in the parentheses denote the sample size. Right: Testing loss of VRLR for different methods under multiple sample sizes.

**Empirical results** We plot the training loss instead of the testing loss since we are comparing different objective functions. Figure 6, 7, and 8 show the empirical results in this part. Note that all the observations in Section 6 also hold: (1) coreset sampling and uniform sampling can drastically reduce the communication complexity where nearly maintain the solution performance, and (2) coreset performs better than uniform sampling under the same number of communication.

### A.3  Different number of centers for VKMC

In this section we test our methods on VKMC using different number of centers.

**Empirical setup** The experimental setup in this part is the same as the setup in Section 6 for VKMC, except that we are using 5 centers instead of 10 centers.

**Empirical results** Figure 9 summarizes the result. All the observations in Section 6 also hold.

### A.4  Experiments on other datasets

In this section, we present the experiment results on another dataset. We choose the `KC House` Dataset [35] for both VRLR and VKMC.

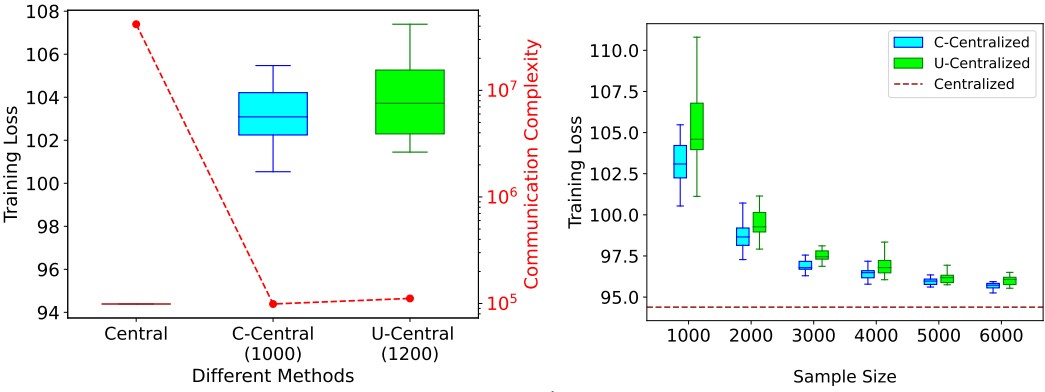

Figure 7: *Results for Lasso regression (Section A.2)* Left: Training loss and communication complexity of VRLR for different methods. C and U means using coreset or uniform sampling. The number in the parentheses denote the sample size. Right: Testing loss of VRLR for different methods under multiple sample sizes.

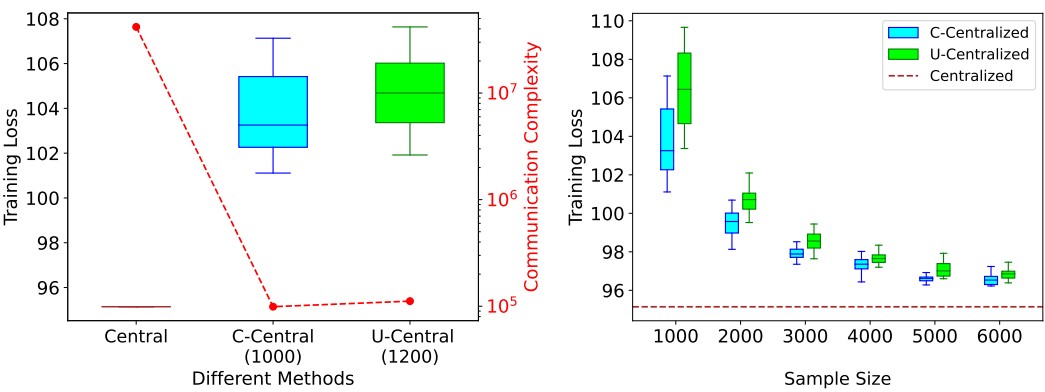

Figure 8: *Results for elastic net (Section A.2)* Left: Training loss and communication complexity of VRLR for different methods. C and U means using coreset or uniform sampling. The number in the parentheses denote the sample size. Right: Testing loss of VRLR for different methods under multiple sample sizes.

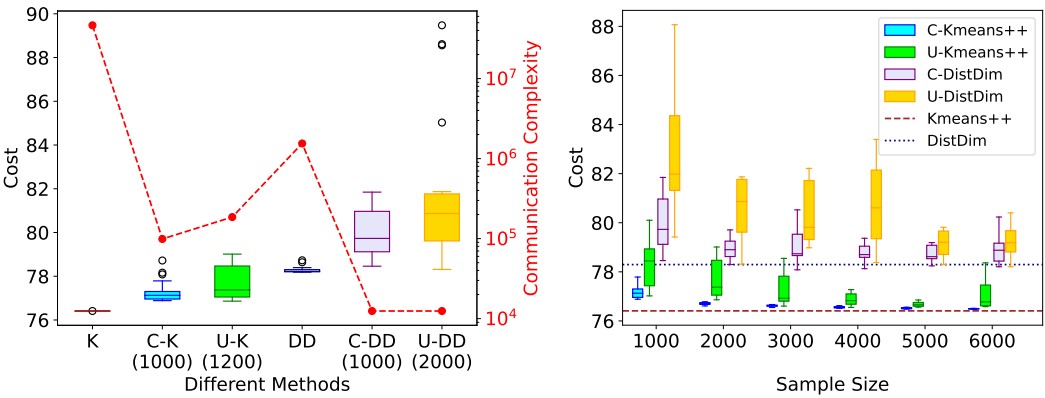

Figure 9: *Results for VKMC with 5 centers (Section A.3)* Left: Cost and communication complexity of VKMC for different methods. C and U means using coreset sampling or uniform sampling. The number in the parentheses denote the sample size. Right: Cost of VKMC for different methods under multiple sample sizes.

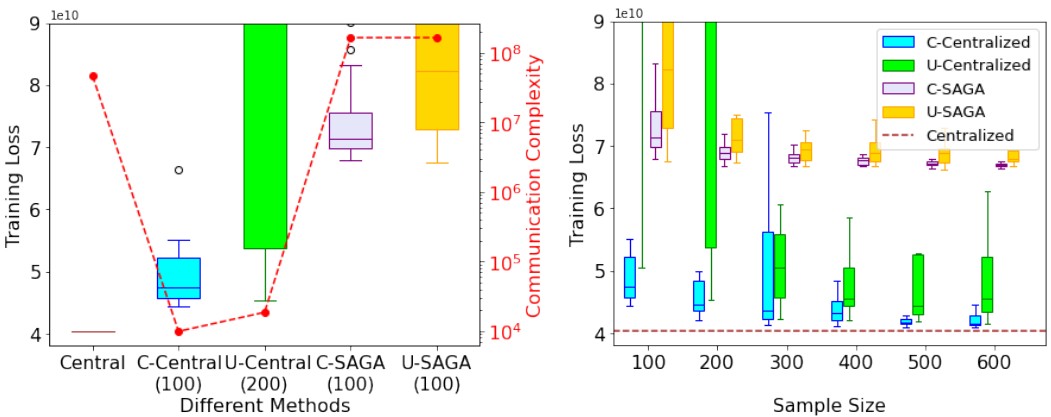

Figure 10: *Results for* KC House *dataset (Section A.4)* Left: Training loss and communication complexity of VRLR for different methods. C and U means using coreset or uniform sampling. The number in the parentheses denote the sample size. Right: Testing loss of VRLR for different methods under multiple sample sizes.

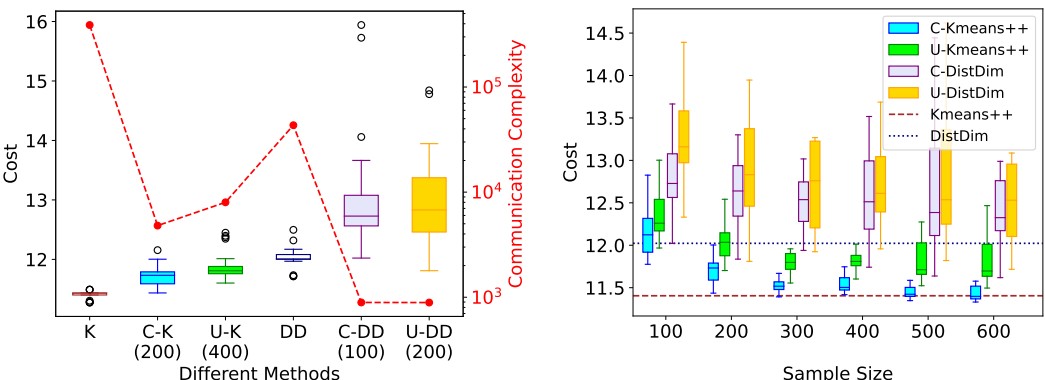

Figure 11: *Results for* KC House *dataset (Section A.4)* Left: Cost and communication complexity of VKMC for different methods. C and U means using coreset sampling or uniform sampling. The number in the parentheses denote the sample size. Right: Cost of VKMC for different methods under multiple sample sizes.

**Empirical setup**    Our experiment setup is nearly the same as the setup in Section 6. However, there are a few differences: (1) the dataset we use is KC House Dataset [35], which contains 21613 data points and each datapoint constains 18 features and a label; (2) we conduct the experiment using only two parties because the limited number of features, we put the first nine features on the first party and the remaining on the second; and (3), we do not consider regularizer for VRLR (plain linear regression). Also note that similar to Section 6, we normalize each feature to have standard deviation 1 during the clustering task.

**Empirical results**    For VRLR, we plot the training loss instead of the testing loss, since the dataset is not so large and coreset does not have theoretical guarantee for generalization error. Figure 10 and 11 summarize our results for VRLR and VKMC respectively. From the results, we still find that our coreset construction method can outperform uniform sampling, and both of them can drastically reduce the communication complexity compared with the original baselines.

Note that in Figure 10, C-SAGA and U-SAGA performs much worse than the baseline Central. However, C-Central can perform much better and has similar performance as Central, and this phenomenon may attribute to the fact that this problem is hard to solve by SAGA algorithm, and using other second-order methods [73] may help. Also note that when the size is small (100 and 200), U-Central may produce "ridiculous" solutions and the cost blows up.

# B  Justification of Data Assumptions

In this section, we justify our data assumptions in Section 4 (Assumption 4.1) and Section 5 (Assumption 5.1). We show that in the *smoothed analysis* regime, Assumption 4.1 and 5.1 are easy to satisfy with some standard assumptions. In Section B.1, we show the results related to Assumption 4.1, and in Section B.2, we justify Assumption 5.1.

## B.1  Justification of Assumption 4.1

In this section, we interpret and justify Assumption 4.1. First, we recall Assumption 4.1.

**Assumption 4.1.** *Let $U^{(j)} \in \mathbb{R}^{n \times d'_j}$ denote the orthonormal basis of the column space of $X^{(j)}$ stored on party $j$ ($U^{(T)}$ denotes the orthonormal basis of $[X^{(T)}, y]$), and then the matrix $U = [U^{(1)}, U^{(2)}, \ldots, U^{(T)}]$ has smallest singular value $\sigma_{\min}(U) \geq \gamma > 0$.*

Assumption 4.1 requires that the subspace generated by any party cannot be included in the subspace generated by all other parties. However, it it not sure what standard assumptions can lead to Assumption 4.1. The following lemma shows that, $\sigma_{\min}(U)$ can be lower bounded by th smallest and largest singular value of matrix $X' = [X, y]$.

**Lemma B.1.** *If matrix $X' = [X, y]$ has smallest singular value $\sigma_{\min}(X') > 0$ and largest singular value $\sigma_{\max}(X')$, we have*

$$\sigma_{\min}(U) \geq \frac{\sigma_{\min}(X')}{\sigma_{\max}(X')}.$$

*Proof.* Because we assume $X'$ has smallest singular value, we can represent $X' = UA$, where $A$ is a $d+1$ by $d+1$ matrix with rank $d+1$.

Now for any $w$, we have

$$\|Uw\| = \|X'A^{-1}w\| \geq \sigma_{\min}(X') \|A^{-1}w\|.$$

Note that $A$ has rank $d+1$, and thus $\sigma_{\min}(A^{-1}) = 1/\sigma_{\max}(A)$. Besides, $A = \mathrm{diag}(A^{(1)}, \ldots, A^{(T)})$ is a block diagonal matrix, where $X^{(j)} = A^{(j)}U^{(j)}$ for $j \in [T-1]$ and $[X^{(T)}, y] = A^{(T)}U^{(T)}$, and thus $\sigma_{\max}(A) = \max_{j \in [T]}\{\sigma_{\max}(A^{(j)})\}$. Because $U^{(j)}$ is the orthonormal basis of $X^{(j)}$ or $[X^{(T)}, y]$, we have

$$\sigma_{\max}(A^{(j)}) = \sigma_{\max}(X^{(j)}), \quad \sigma_{\max}(A^{(T)}) = \sigma_{\max}([X^{(T)}, y]).$$

We also have $\sigma_{\max}(X') \geq \sigma_{\max}(X^{(j)})$ and $\sigma_{\max}(X') \geq \sigma_{\max}([X^{(T)}, y])$. Combining all the properties together, we get $\sigma_{\max}(A) \leq \sigma_{\max}(X')$, and thus conclude the proof. $\square$

Using the preivous lemma, it is easy to analyze the smallest singular value $\sigma(U)$ in the smoothed analysis regime. Specifically, we prove that for any dataset $[X, y]$ satisfying certain conditions, we add a random perturbation on the dataset, resulting $[X_p, y_p]$, and we show that with high probability, $U_p$ (which is constructed from dataset $[X_p, y_p]$ has smallest singular value. The result is formalized in the following theorem.

**Theorem B.1.** *There exists constant $n_0$ such that for any dataset $[X, y] \in \mathbb{R}^{n \times (d+1)}$ where each data point $\|[x_i; y_i]\|_2^2 \leq B$ and $n \geq 2d, n \geq n_0$. If we perturb the dataset by a small random Gaussian noise $[X_p, y_p]$ where $X_p = X + Z$, $y_p = y + w$, and each coordinate of $Z$ and $w$ comes from $\mathcal{N}(0, r^2 B^2)$, then with high probability, the basis $U_p$ computed from $[X_p, y_p]$ has smallest singular value at least $\Omega(r)$.*

In order to prove Theorem B.1, we use the following theorem (Theorem 1.1 in [8]).

**Proposition B.1** (Smoothed analysis of condition number, Theorem 1.1 in [8])**.** *Suppose that $\bar{A} \in \mathbb{R}^{n \times d}$ satisfies $\|\bar{A}\| \leq 1$, and let $0 < r_p \leq 1$. Then,*

$$\Pr_{A \sim \mathcal{N}(\bar{A}, r^2 I)} \{\kappa(A) \geq C_1 t\} \leq \left(C_2/t + C_2/r_p \sqrt{n}t\right)^{n-d+1},$$

*for some constants $C_1, C_2, C_3$ and all $t \geq C_3$.*

Roughly speaking, Proposition B.1 claims that with high probability, the condition number under the smoothed analysis regime should be bounded above. Then with the help of Lemma B.1 and Proposition B.1, we can now prove Theorem B.1.

*Proof of Theorem B.1.* For simplicity, we treat denote $\boldsymbol{D} = [\boldsymbol{X}, \boldsymbol{y}]$ and $\boldsymbol{D}_p = [\boldsymbol{X}_p, \boldsymbol{y}_p]$, and $\boldsymbol{D}_p = \boldsymbol{D} + \boldsymbol{A}$, where each coordinate of $\boldsymbol{A}$ comes form $\mathcal{N}(0, r^2 B^2)$.

Note that the condition number of a matrix is 'scale invariant', which means that

$$\kappa(\boldsymbol{A}) = \kappa(c\boldsymbol{A}),$$

for constants $c \neq 0$.

Now, since the row of $\boldsymbol{D}$ has bounded norm $B$, thus $\|\boldsymbol{D}\| \leq B\sqrt{n}$. By the scale invariance of condition number, we have

$$\kappa(\boldsymbol{D}_p) = \kappa(\boldsymbol{D}_p/(B\sqrt{n})).$$

Now, the perturbation factor $r_p$ in Proposition B.1 is $rB/B\sqrt{n} = r/\sqrt{n}$, and we know that

$$\Pr\{\kappa(\boldsymbol{D}_p) \geq C_1/r\} \leq (C_2 r + C_2)^{n-d+1},$$

for some constants $C_1, C_3 > 0$, constant $C_2$ s.t. $0 < C_2 < 1$ and all $r \leq C_3$. Directly applying Lemma B.1 concludes the proof. $\square$

## B.2 Justification of Assumption 5.1

In this section, we justify Assumption 5.1. We first recall the assumption.

**Assumption 5.1.** *There exists $\tau \geq 1$ and some party $t \in [T]$ such that $\|\boldsymbol{x}_i - \boldsymbol{x}_j\|^2 \leq \tau \left\|\boldsymbol{x}_i^{(t)} - \boldsymbol{x}_j^{(t)}\right\|^2$ for any $i, j \in [n]$.*

Roughly speaking, this assumption requires there is a party that is "important", and any two data points which can be differentiated can also be differentiated on that party to some extent. In reality, this assumption should be approximately satisfied since different features should be "correlated".

Next, similar to the justification of Assumption 4.1, we use smoothed analysis framework to show that for dataset $\boldsymbol{X}$ under certain conditions, by perturbing the dataset for a little bit, Assumption 5.1 will be satisfied with high probability. Formally, we have the following theorem.

**Theorem B.2.** *For any dataset where each data point $\|\boldsymbol{x}_i\|_2^2 \leq B$ for all $\boldsymbol{x}_i \in \boldsymbol{X}$ and $\max_{j \in [T]} d_j \geq \Omega(\log^2 n)$. If we perturb the dataset by a small random Gaussian noise $\boldsymbol{X}_p$ where $\boldsymbol{X}_p = \boldsymbol{X} + \boldsymbol{Z}$, and each coordinate of $\boldsymbol{Z}$ and $\boldsymbol{w}$ comes from $\mathcal{N}(0, r^2 B^2)$. Then with high probability, $\boldsymbol{X}_p$ satisfies Assumption 5.1 with*

$$\tau = O\left(\frac{1}{r^2} + \frac{d}{\log^2 n}\right)$$

The intuition of the proof is that, the norm of a high-dimensional (sub-)gaussian random vector should concentrate around $\Theta(\sqrt{d})$, where $d$ is the dimension of the (sub-)gaussian random vector. Thus, as long as we add some perturbation to the original dataset, the norm of the difference between any two perturbed data points on party $j$ should be at least $\sqrt{d_j}$. Formally, we have the following proposition for the concentration of norm.

**Proposition B.2** (Concentration of the norm). *Let $\boldsymbol{\xi} = (\boldsymbol{\xi}_1, \ldots, \boldsymbol{\xi}_d) \in \mathbb{R}^d$ be a random Gaussian vector, where each coordinate is sampled from $\mathcal{N}(0, r^2)$ independently. Then there exists constants $c$ such that for any $t \geq 0$,*

$$\Pr\left\{\left|\|\boldsymbol{\xi}\|_2 - r\sqrt{d}\right| \geq rt\right\} \leq 2\exp\left(-ct^2\right)$$

Now with the help of this proposition, we can now prove Theorem B.2.

*Proof of Theorem B.2.* First, we upper bound $\|\tilde{\boldsymbol{x}}_i - \tilde{\boldsymbol{x}}_j\|^2$ where $\tilde{\boldsymbol{x}}_i$ denote the $i$-th perturbed data and we use $\boldsymbol{\xi}_i = \tilde{\boldsymbol{x}}_i - \boldsymbol{x}_i$ to denote the random perturbation. We have

$$\|\tilde{\boldsymbol{x}}_i - \tilde{\boldsymbol{x}}_j\|^2 \leq 2\|\boldsymbol{x}_i - \boldsymbol{x}_j\|^2 + 2\|\boldsymbol{\xi}_i - \boldsymbol{\xi}_j\|^2.$$

From the assumption, we know that $\|\boldsymbol{x}_i - \boldsymbol{x}_j\| \leq 2B$, and thus we only need to bound the second term. From Proposition B.2, we know that for fixed $i \neq j$, we have

$$\Pr\left\{\left|\|\boldsymbol{\xi}_i - \boldsymbol{\xi}_j\| - \sqrt{2}rB\sqrt{d}\right| \geq crB\log n\right\} \leq 2\exp\left(4\log n\right),$$

for some constants $c$ since $\boldsymbol{\xi}_i - \boldsymbol{\xi}_j$ is a Gaussian random vector whose entries are drawn from $\mathcal{N}(0, 2r^2B^2)$. Thus, with probability at least $1 - 2/n^4$, we have

$$\|\tilde{\boldsymbol{x}}_i - \tilde{\boldsymbol{x}}_j\|^2 \leq 8B^2 + 4r^2B^2d + cr^2B^2\log^2 n,$$

for some constant $c$. Then applying the union bound, we know that with probability at least $1 - \frac{1}{n^2}$,

$$\|\tilde{\boldsymbol{x}}_i - \tilde{\boldsymbol{x}}_j\|^2 \leq 8B^2 + cr^2B^2\log^2 n, \forall i \neq j,$$

for some constant $c$. Without loss of generality, suppose that $d_1 = \max_{j\in[T]} d_j$, and then we lower bound $\left\|\tilde{\boldsymbol{x}}_i^{(1)} - \tilde{\boldsymbol{x}}_j^{(1)}\right\|^2$. First since $\left\|\tilde{\boldsymbol{x}}_i^{(1)} - \tilde{\boldsymbol{x}}_j^{(1)}\right\|^2$ is the noncentralized $\chi^2$ distribution, we have

$$\Pr\left\{\left\|\tilde{\boldsymbol{x}}_i^{(1)} - \tilde{\boldsymbol{x}}_j^{(1)}\right\|^2 \geq t\right\} \geq \Pr\left\{\left\|\boldsymbol{\xi}_i^{(1)} - \boldsymbol{\xi}_j^{(1)}\right\|^2 \geq t\right\}.$$

Then from Proposition B.2, we have

$$\Pr\left\{\left|\left\|\boldsymbol{\xi}_i^{(1)} - \boldsymbol{\xi}_j^{(1)}\right\| - \sqrt{2}rB\sqrt{d_1}\right| \geq crB\log n\right\} \leq 2\exp\left(4\log n\right),$$

for some constant $c$. Thus, if $d_1 \geq C\log^2 n$ for some large enough constant $c$, we know that with probability at least $1 - 2/n^4$,

$$\left\|\tilde{\boldsymbol{x}}_i^{(1)} - \tilde{\boldsymbol{x}}_j^{(1)}\right\|^2 \geq cr^2B^2\log^2 n,$$

for some constant $c$. Then with a union bound, we know that with probability at least $1 - 1/n^2$,

$$\left\|\tilde{\boldsymbol{x}}_i^{(1)} - \tilde{\boldsymbol{x}}_j^{(1)}\right\|^2 \geq cr^2B^2\log^2 n, \forall i \neq j,$$

for some constant $c$. Combining with the previous part, we know that if $\max_{j\in[T]} d_j \geq C\log^2 n$ for some large constant $C$, then with probability at least $1 - \frac{1}{n}$, we have

$$\frac{\|\tilde{\boldsymbol{x}}_i - \tilde{\boldsymbol{x}}_j\|^2}{\left\|\tilde{\boldsymbol{x}}_i^{(1)} - \tilde{\boldsymbol{x}}_j^{(1)}\right\|^2} \leq O\left(\frac{B^2 + r^2B^2d + r^2B^2\log^2 n}{r^2B^2\log^2 n}\right) = O\left(\frac{1}{r^2} + \frac{d}{\log^2 n}\right).$$

$\square$

## C   Proof of Theorem 2.5

*Proof of Theorem 2.5.* We only take VRLR as an example. We consider the following communication scheme: First apply the communication scheme $A'$ to construct an $\varepsilon$-coreset $(S, w)$ for VRLR in the server; then the server broadcasts $(S, w)$ to all parties; and finally apply the communication scheme $A$ to $(S, w)$ and obtain a solution $\boldsymbol{\theta} \in \mathbb{R}^d$ in the server.

Let $\boldsymbol{\theta}^\star$ be the optimal solution for the offline regularized linear regression problem.

By the coreset definition, we have that

$$
\begin{array}{rll}
\mathsf{cost}^R(X, \boldsymbol{\theta}) \leq & (1+\varepsilon)\mathsf{cost}^R(S, \boldsymbol{\theta}) & \text{(by coreset definition)} \\
\leq & (1+\varepsilon)\alpha \cdot \mathsf{cost}^R(S, \boldsymbol{\theta}^\star) & \text{(by } A\text{)} \\
\leq & (1+\varepsilon)^2\alpha \cdot \mathsf{cost}^R(X, \boldsymbol{\theta}^\star) & \text{(by coreset definition)} \\
\leq & (1+3\varepsilon)\alpha \cdot \mathsf{cost}^R(X, \boldsymbol{\theta}^\star), & (\varepsilon \in (0,1))
\end{array}
$$

which proves the approximation ratio.

For the total communication complexity, note that the broadcasting step costs $2Tm$. This completes the proof. $\square$

# D Proof of Theorem 3.1

For preparation, we first introduce a well-known importance sampling framework for offline coreset construction by [22, 7].

**Theorem D.1** (**Feldman-Langberg framework [22, 7]**). *Let $\varepsilon, \delta \in (0, 1/2)$ and let $k \geq 1$ be an integer. Let $\boldsymbol{X} \subset \mathbb{R}^d$ be a dataset of $n$ points together with a label vector $\boldsymbol{y} \in \mathbb{R}^n$, and $\boldsymbol{g} \in \mathbb{R}_{\geq 0}^n$ be a vector. Let $\mathcal{G} := \sum_{i \in [n]} g_i$. Let $S \subseteq [n]$ be constructed by taking $m \geq 1$ samples, where each sample $i \in [n]$ is selected with probability $\frac{g_i}{\mathcal{G}}$ and has weight $w(i) := \frac{\mathcal{G}}{|S|g_i}$. Then we have*

- *If $g_i \geq \sup_{\boldsymbol{\theta} \in \mathbb{R}^d} \frac{\text{cost}_i^R(\boldsymbol{X}, \boldsymbol{\theta})}{\text{cost}^R(\boldsymbol{X}, \boldsymbol{\theta})}$ holds for any $i \in [n]$ and $m = O\left(\varepsilon^{-2} \mathcal{G}(d^2 \log \mathcal{G} + \log(1/\delta))\right)$, with probability at least $1 - \delta$, $(S, w)$ is an $\varepsilon$-coreset for offline regularized linear regression.*

- *If $g_i \geq \sup_{\boldsymbol{C} \in \mathcal{C}} \frac{\text{cost}_i^C(\boldsymbol{X}, \boldsymbol{C})}{\text{cost}^C(\boldsymbol{X}, \boldsymbol{C})}$ holds for any $i \in [n]$ and $m = O\left(\varepsilon^{-2} \mathcal{G}(dk \log \mathcal{G} + \log(1/\delta))\right)$, with probability at least $1 - \delta$, $(S, w)$ is an $\varepsilon$-coreset for offline $k$-means clustering.*

We call $g_i$ the sensitivity of point $\boldsymbol{x}_i$ that represents the maximum contribution of $\boldsymbol{x}_i$ over all possible parameters, and call $\mathcal{G}$ the total sensitivity. By [65], we note that the total sensitivity can be upper bounded by $O(d)$ for offline regularized linear regression and by $O(k)$ for offline $k$-means clustering. By the Feldman-Langberg framework, it suffices to compute a sensitivity vector $\boldsymbol{g} \in \mathbb{R}^n$ for offline coreset construction.

*Proof of Theorem 3.1.* We first discuss the communication complexity of Algorithm 1. At the first round, the communication complexity in Line 2 is $T$ and in Line 4 is $T$. At the second round, the communication complexity in Line 5 is at most $\sum_{j \in [T]} a_j = m$ and in Line 6 is at most $mT$. At the third round, the communication complexity in Line 7 is at most $mT$. Overall, the total communication complexity is $O(mT)$.

Next, we prove the correctness. We only take VRLR as an example and the proof for VKMC is similar. Note that each sample in $S$ is equivalent to be drawn by the following procedure: Sample $i \in [n]$ with probability $\sum_{j \in [T]} g_i^{(j)} / \mathcal{G}$. This is because by Lines 3 and 5, the sampling probability of $i \in [n]$ is exactly

$$\sum_{j \in [T]} \frac{\mathcal{G}^{(j)}}{\mathcal{G}} \cdot \frac{g_i^{(j)}}{\mathcal{G}^{(j)}} = \frac{\sum_{j \in [T]} g_i^{(j)}}{\mathcal{G}}.$$

Then letting $g_i' = \zeta \cdot \sum_{j \in [T]} g_i^{(j)}$ for each $i \in [n]$, we have

$$g_i' \geq \sup_{\boldsymbol{\theta} \in \mathbb{R}^d} \frac{\text{cost}_i^R(\boldsymbol{X}, \boldsymbol{\theta})}{\text{cost}^R(\boldsymbol{X}, \boldsymbol{\theta})}$$

by assumption. This completes the proof by plugging $g_i'$ to Theorem D.1. $\qquad\square$

# E Omitted Proof in Section 4

## E.1 Communication lower bound for VRLR coreset construction

The proof is via a reduction from an EQUALITY problem to the problem of coreset construction for VRLR. For preparation, we first introduce some concepts in the field of communication complexity.

**Communication complexity.** Here it suffices to consider the two-party case ($T = 2$). Assume we have two players Alice and Bob, whose inputs are $x \in \mathcal{X}$ and $y \in \mathcal{Y}$ respectively. They exchange messages with a coordinator according to a protocol $\Pi$ (deterministic/randomized) to compute some function $f : \mathcal{X} \times \mathcal{Y} \to \mathcal{Z}$. For the input $(x, y)$, the coordinator outputs $\Pi(x, y)$ when Alice and Bob run $\Pi$ on it. We also use $\Pi(x, y)$ to denote the transcript (concatenation of messages). Let $|\Pi_{x,y}|$ be the length of the transcript. The communication complexity of $\Pi$ is defined as $\max_{x,y} |\Pi_{x,y}|$. If $\Pi$ is a randomized protocol, we define the *error* of $\Pi$ by $\max_{x,y} \mathbb{P}(\Pi(x, y) \neq f(x, y))$, where the max is over all inputs $(x, y)$ and the probability is over the randomness used in $\Pi$. The $\delta$-*error randomized communication complexity* of $f$, denoted by $R_\delta(f)$, is the minimum communication complexity of any protocol with error at most $\delta$.

**EQUALITY problem.** In the EQUALITY problem, Alice holds $a = \{a_1, \ldots, a_n\} \in \{0, 1\}^n$ and Bob holds $b = \{b_1, \ldots, b_n\} \in \{0, 1\}^n$. The goal is to compute EQUALITY$(a, b)$ which equals 1 if $a_i = b_i$ for all $i \in [n]$ otherwise 0. The following lemma gives a well-known lower bound for deterministic communication protocols that correctly compute EQUALITY function.

**Lemma E.1** (**Communication complexity of EQUALITY [42]**). *The deterministic communication complexity of EQUALITY is* $\Omega(n)$.

**Reduction from EQUALITY.** Now we are ready to prove Theorem 4.1.

*Proof of Theorem 4.1.* We prove this by a reduction from EQUALITY. For simplicity, it suffices to assume $d = 1$ and $T = 2$ in the VRLR problem. Given an EQUALITY instance of size $n$, let $a \in \{0, 1\}^n$ be Alice's input and $b \in \{0, 1\}^n$ be Bob's input. They construct inputs $\boldsymbol{X} \in \mathbb{R}^n$ and $\boldsymbol{y} \in \mathbb{R}^n$ for VRLR, where $\boldsymbol{X} = a$ and $\boldsymbol{y} = b$. We denote $S \subseteq [n]$ with a weight function $w : S \to \mathbb{R}_{\geq 0}$ to be an $\varepsilon$-coreset such that for any $\boldsymbol{\theta} \in \mathbb{R}$, we have

$$\mathsf{cost}^R(S, \boldsymbol{\theta}) := \sum_{i \in S} w(i) \cdot (\boldsymbol{x}_i^\top \boldsymbol{\theta} - y_i)^2 + R(\boldsymbol{\theta}) \in (1 \pm \varepsilon) \cdot \mathsf{cost}^R(\boldsymbol{X}, \boldsymbol{\theta}).$$

Based on the above guarantee, w.l.o.g, if we set $\theta = 1$ and $R = 0$, then there exist two cases with positive cost: $(a_i, b_i) = (0, 1)$ or $(1, 0)$. In other words, EQUALITY$(a, b) = 0$ if and only if the set $\{(\boldsymbol{x}_i, y_i) : i \in S\}$ includes $(0, 1)$ or $(1, 0)$. Thus, any deterministic protocol for VRLR coreset construction can be used as a deterministic protocol for EQUALITY. The lower bound follows from Lemma E.1. □

## E.2 Proof of Theorem 4.2

In this section, we show the detailed proof of Theoem 4.2. The proof idea is to bound the sensitivity of each data point and then apply Theorem 3.1. Recall that in Theorem 3.1, we define

$$\zeta = \max_{i \in [n]} \sup_{\boldsymbol{\theta} \in \mathbb{R}^d} \frac{\mathsf{cost}_i^R(\boldsymbol{X}, \boldsymbol{\theta})}{\mathsf{cost}^R(\boldsymbol{X}, \boldsymbol{\theta})} \Big/ \sum_{j \in [T]} g_i^{(j)}.$$

We first show the following main lemma.

**Lemma E.2.** *Under Assumption 4.1, the sensitivity of a data point can be bounded by*

$$\sup_{\boldsymbol{\theta} \in \mathbb{R}^d} \frac{\mathsf{cost}_i^R(\boldsymbol{X}, \boldsymbol{\theta})}{\mathsf{cost}^R(\boldsymbol{X}, \boldsymbol{\theta})} \leq \frac{g_i}{\gamma^2},$$

*which means that* $\zeta \leq 1/\gamma^2$.

*Proof.* The sensitivity function for each data point $(\boldsymbol{x}_i, y_i)$ is defined as

$$\sup_{\boldsymbol{\theta} \in \mathbb{R}^d} \frac{\mathsf{cost}_i^R(\boldsymbol{X}, \boldsymbol{\theta})}{\mathsf{cost}^R(\boldsymbol{X}, \boldsymbol{\theta})} = \sup_{\boldsymbol{\theta}} \frac{(\boldsymbol{x}_i^\top \boldsymbol{\theta} - y_i)^2 + \frac{\lambda R(\boldsymbol{\theta})}{n}}{\|\boldsymbol{X}\boldsymbol{\theta} - \boldsymbol{y}\|^2 + \lambda R(\boldsymbol{\theta})}.$$

First, we have

$$\sup_{\boldsymbol{\theta} \in \mathbb{R}^d} \frac{(\boldsymbol{x}_i^\top \boldsymbol{\theta} - y_i)^2 + \frac{\lambda R(\boldsymbol{\theta})}{n}}{\|\boldsymbol{X}\boldsymbol{\theta} - \boldsymbol{y}\|^2 + \lambda R(\boldsymbol{\theta})} = \sup_{\boldsymbol{\theta} \in \mathbb{R}^d} \left( \frac{(\boldsymbol{x}_i^\top \boldsymbol{\theta} - y_i)^2}{\|\boldsymbol{X}\boldsymbol{\theta} - \boldsymbol{y}\|^2 + \lambda R(\boldsymbol{\theta})} + \frac{\frac{\lambda R(\boldsymbol{\theta})}{n}}{\|\boldsymbol{X}\boldsymbol{\theta} - \boldsymbol{y}\|^2 + \lambda R(\boldsymbol{\theta})} \right)$$

$$\leq \sup_{\boldsymbol{\theta} \in \mathbb{R}^d} \left( \frac{(\boldsymbol{x}_i^\top \boldsymbol{\theta} - y_i)^2}{\|\boldsymbol{X}\boldsymbol{\theta} - \boldsymbol{y}\|^2} + \frac{1}{n} \right),$$

where we separate the regression loss and the regularized loss.

Then for the regression loss, define $\boldsymbol{X}' = [\boldsymbol{X}, \boldsymbol{y}]$ and $d' = \sum_{j \in [T]} d_j'$, we have

$$\sup_{\boldsymbol{\theta} \in \mathbb{R}^d} \frac{(\boldsymbol{x}_i^\top \boldsymbol{\theta} - y_i)^2}{\|\boldsymbol{X}\boldsymbol{\theta} - \boldsymbol{y}\|^2} \leq \sup_{\boldsymbol{\theta} \in \mathbb{R}^{d+1}} \frac{((\boldsymbol{x}_i')^\top \boldsymbol{\theta})^2}{\|\boldsymbol{X}'\boldsymbol{\theta}\|^2} = \sup_{\boldsymbol{\theta} \in \mathbb{R}^{d'}} \frac{((\boldsymbol{u}_i)^\top \boldsymbol{\theta})^2}{\|\boldsymbol{U}\boldsymbol{\theta}\|^2}$$

Note that under Assumption 4.1, matrix $\boldsymbol{U}$ has smallest singular value $\sigma_{\min} \geq \gamma > 0$, and we can get

$$
\sup_{\boldsymbol{\theta} \in \mathbb{R}^d} \frac{(\boldsymbol{x}_i^\top \boldsymbol{\theta} - y_i)^2}{\|\boldsymbol{X}\boldsymbol{\theta} - \boldsymbol{y}\|^2} \leq \sup_{\boldsymbol{\theta} \in \mathbb{R}^{d'}} \frac{((\boldsymbol{u}_i)^\top \boldsymbol{\theta})^2}{\|\boldsymbol{U}\boldsymbol{\theta}\|^2}
$$

$$
\leq \sup_{\boldsymbol{\theta} \in \mathbb{R}^{d'}} \frac{((\boldsymbol{u}_i)^\top \boldsymbol{\theta})^2}{\sigma_{\min}^2 \|\boldsymbol{\theta}\|^2}
$$

$$
\leq \frac{\|\boldsymbol{u}_i\|^2}{\gamma^2}
$$

$$
= \frac{\sum_{j \in [T]} \left\|\boldsymbol{u}_i^{(j)}\right\|^2}{\gamma^2}.
$$

Recall that $g_i = \sum_{j \in [T]} g_i^{(j)} = \sum_{j \in [T]} \left\|\boldsymbol{u}_i^{(j)}\right\|^2 + \frac{T}{n}$. Hence,

$$
\sup_{\boldsymbol{\theta} \in \mathbb{R}^d} \frac{\mathrm{cost}_i^R(\boldsymbol{X}, \boldsymbol{\theta})}{\mathrm{cost}^R(\boldsymbol{X}, \boldsymbol{\theta})} \leq \frac{\sum_{j \in [T]} \left\|\boldsymbol{u}_i^{(j)}\right\|^2}{\gamma^2} + \frac{1}{n} \leq \frac{g_i}{\gamma^2}.
$$

$\square$

Now with the help of Lemma E.2, we can prove Theorem 4.2.

*Proof of Theorem 4.2.* Note that from Lemma E.2, we know that $\zeta \leq 1/\gamma^2$. Also note that from Algorithm 2, we have

$$
\mathcal{G} = \sum_{j \in [T]} \sum_{i \in [n]} \left( \left\|\boldsymbol{u}_i^{(j)}\right\|^2 + \frac{1}{n} \right) = \sum_{j \in [T]} \left\|\boldsymbol{U}^{(j)}\right\|_{\mathrm{F}}^2 + T = \sum_{j \in [T]} d_j' + T \leq d + T + 1 \leq 2d + 1.
$$

Then we apply Theorem 3.1, the $\varepsilon$-coreset size for VRLR can be bounded by

$$
m = O(\varepsilon^{-2} \gamma^{-2} d(d^2 \log(\gamma^{-2} d) + \log 1/\delta)),
$$

and the communication complexity is $O(mT)$.

$\square$

# F Omitted Proof in Section 5

## F.1 Communication lower bound for VKMC coreset construction

The proof is via a reduction from a set-disjointness (DISJ) problem to the problem of coreset construction for VKMC.

**DISJ problem.** In the DISJ problem, Alice holds $a = \{a_1, \ldots, a_n\} \in \{0, 1\}^n$ and Bob holds $b = \{b_1, \ldots, b_n\} \in \{0, 1\}^n$. The goal is to compute $\mathrm{DISJ}(a, b) = \bigvee_{i \in [n]} (a_i \bigwedge b_i)$. The following lemma gives a well-known communication lower bound for DISJ.

**Lemma F.1 (Communication complexity of DISJ [37, 59, 3]).** *The randomized communication complexity of DISJ is $\Omega(n)$, i.e., for $\delta \in [0, 1/2)$ and $n \geq 1$, $R_\delta(DISJ) = \Omega(n)$.*

**Reduction from DISJ.** Now we are ready to prove Theorem 5.1.

*Proof of Theorem 5.1.* We prove this by a reduction from DISJ. For simplicity, it suffices to assume $d = 2$ and $T = 2$ in the VKMC problem. Given a DISJ instance of size $n$, let $a \in \{0, 1\}^n$ be Alice's input and $b \in \{0, 1\}^n$ be Bob's input. They construct an input $\boldsymbol{X} \subset \mathbb{R}^2$ for VKMC, where $\boldsymbol{X} = \{\boldsymbol{x}_i : \boldsymbol{x}_i = (a_i, b_i), i \in [n]\}$. We denote $S \subseteq [n]$ with a weight function $w : S \to \mathbb{R}_{\geq 0}$ to be an $\varepsilon$-coreset such that for any $\boldsymbol{C} \in \mathcal{C}$ with $|\boldsymbol{C}| = k$, we have

$$
\mathrm{cost}^C(S, \boldsymbol{C}) := \sum_{i \in S} w(i) \cdot d(\boldsymbol{x}_i, \boldsymbol{C})^2 \in (1 \pm \varepsilon) \cdot \mathrm{cost}^C(\boldsymbol{X}, \boldsymbol{C}).
$$

Based on the above guarantee, w.l.o.g., if we set $k = 3$ and $C = \{(0,0),(0,1),(1,0)\}$, then only point $(1,1)$ can induce positive cost. In other words, $\text{DISJ}(a,b) = 1$ if and only if the set $\{x_i : i \in S\}$ includes point $(1,1)$. Thus, any $\delta$-error protocol for VKMC coreset construction can be used as a $\delta$-error protocol for DISJ. The lower bound follows from Lemma F.1. $\qquad\square$

## F.2 Proof of Theorem 5.2

Algorithm 3 applies the meta Algorithm 1 after computing $\{g_i^{(j)}\}$ locally. The key is to construct local sensitivities $g_i^{(j)}$ so that the sum $\sum_{j \in [T]} g_i^{(j)}$ can approximate global sensitivity $g_i$ well, i.e, with both small $\zeta$ and $\mathcal{G}$ in Theorem 3.1.

**Constructing local sensitivities.** By the local sensitivities $g_i^{(j)}$ defined in Line 10 of Algorithm 3, we have the following lemma that upper bound both $\zeta$ and $\mathcal{G}$.

**Lemma F.2** (**Upper bounding the global sensitivity of VKMC locally**). *Given a dataset $\boldsymbol{X} \subset \mathbb{R}^d$ with Assumption 5.1, an $\alpha$-approximation algorithm for $k$-means with $\alpha = O(1)$ and integers $k \geq 1$, $T \geq 1$, the local sensitivities $g_i^{(j)}$ in Algorithm 3 satisfies that for any $i \in [n]$, $\sup_{\boldsymbol{C} \in \mathcal{C}} \frac{\text{cost}_i^C(\boldsymbol{X}, \boldsymbol{C})}{\text{cost}^C(\boldsymbol{X}, \boldsymbol{C})} \leq 4\tau \sum_{j \in [T]} g_i^{(j)}$, i.e., $\zeta = O(\tau)$. Moreover, $\mathcal{G} := \sum_{i \in [n], j \in [T]} g_i^{(j)} = O(\alpha kT)$.*

The proof can be found in Section F.3, and it is partly modified from the dimension-reduction type argument [65], which upper bounds the total sensitivity of a point set in clustering problem by projecting points onto an optimal solution. Intuitively, if some party $t$ satisfies Assumption 5.1, the partition over $[n]$ corresponding to an $\alpha$-approximation computed using local data will induce a global $\alpha\tau$-approximate solution. Hence, combining this with the argument mentioned above, we derive that $g_i^{(t)}$ (scaled by $4\tau$) is an upper bound of the global sensitivity. Though unaware of which party satisfies Assumption 5.1, it suffices to sum up $g_i^{(j)}$ over $j \in [T]$, costing an addtional $T$ in $\mathcal{G}$.

*Proof of Theorem 5.2.* By Lemma F.2, the sensitivity gap $\zeta$ is $O(\tau)$ and the total sensitivity $\mathcal{G}$ is $O(\alpha kT)$. Plugging them into Theorem 3.1 completes the proof. $\qquad\square$

## F.3 Proof of Lemma F.2

Our proof is partly inspired by [65]. For preparation, we first introduce the following useful notations.

Suppose the party $t$ in the dataset $\boldsymbol{X}$ satisfies Assumption 5.1, and $\mathcal{A}$ is an $\alpha$-approximation algorithm for $k$-means clustering. Let $\tilde{\boldsymbol{C}}^{(t)}$ be an $\alpha$-approximate solution computed locally in party $t$ using $\mathcal{A}$, i.e., $\tilde{\boldsymbol{C}}^{(t)} = \mathcal{A}(\boldsymbol{X}^{(t)}) = \{\tilde{\boldsymbol{c}}_l^{(t)} : l \in [k]\}$. We define a mapping $\pi : [n] \to [k]$ to find the closest center index for each point in the local solution, i.e., $\pi(i) = \arg\min_{l \in [k]} d(\boldsymbol{x}_i^{(t)}, \tilde{\boldsymbol{c}}_l^{(t)})$. We also denote $\boldsymbol{B}_l^{(t)} := \{i \in [n] : \pi(i) = l\}$ to be the local cluster corresponding to $\tilde{\boldsymbol{c}}_l^{(t)}$. Note that $\{\boldsymbol{B}_l^{(t)} : l \in [k]\}$ is a partition over data as $\boldsymbol{B}_l^{(t)} \cap \boldsymbol{B}_{l'}^{(t)} = \varnothing$ ($l, l' \in [k], l \neq l'$) and $\cup_{l \in [k]} \boldsymbol{B}_l^{(t)} = [n]$. Let $\tilde{\boldsymbol{C}} := \{\tilde{\boldsymbol{c}}_l : \tilde{\boldsymbol{c}}_l = \frac{1}{|\boldsymbol{B}_l^{(t)}|} \sum_{i \in \boldsymbol{B}_l^{(t)}} \boldsymbol{x}_i\}$ be a $k$-center set in $\mathbb{R}^d$ lifted from $\mathbb{R}^{d_t}$ based on $\{\boldsymbol{B}_l^{(t)}\}$. The following lemma shows that $\tilde{\boldsymbol{C}}$ is also a constant approximation to the global $k$-means clustering.

**Lemma F.3** (**Local partition induces global constant apporximation for $k$-means**). *If party $t$ of a dataset $\boldsymbol{X} \subset \mathbb{R}^d$ satisfies Assumption 5.1, then given a local $\alpha$-approximate solution $\tilde{\boldsymbol{C}}^{(t)}$, for any $k$-center set $\boldsymbol{C} \in \mathcal{C}$, we have*

$$\text{cost}^C(\boldsymbol{X}, \tilde{\boldsymbol{C}}) \leq \tau \text{cost}^C(\boldsymbol{X}^{(t)}, \tilde{\boldsymbol{C}}^{(t)}) \leq \alpha\tau \text{cost}^C(\boldsymbol{X}, \boldsymbol{C}).$$

*Thus, $\tilde{\boldsymbol{C}}$ is an $\alpha\tau$-approximate solution to the global $k$-means clustering.*

*Proof.*

$$\text{cost}^C(\boldsymbol{X}, \tilde{\boldsymbol{C}}) = \sum_{i=1}^{n} d(\boldsymbol{x}_i, \tilde{\boldsymbol{C}})^2$$

$$\leq \sum_{l=1}^{k} \sum_{i \in \boldsymbol{B}_l^{(t)}} d(\boldsymbol{x}_i, \tilde{\boldsymbol{c}}_l)^2 \qquad \text{(assignment by } \boldsymbol{B}_l^{(t)} \text{ is not optimal)}$$

$$= \sum_{l=1}^{k} \frac{1}{2|\boldsymbol{B}_l^{(t)}|} \sum_{i,j \in \boldsymbol{B}_l^{(t)}} d(\boldsymbol{x}_i, \boldsymbol{x}_j)^2 \qquad \text{(a standard property of } k\text{-means )}$$

$$\leq \sum_{l=1}^{k} \frac{\tau}{2|\boldsymbol{B}_l^{(t)}|} \sum_{i,j \in \boldsymbol{B}_l^{(t)}} d(\boldsymbol{x}_i^{(t)}, \boldsymbol{x}_j^{(t)})^2 \qquad \text{(by Assumption 5.1)}$$

$$= \tau \mathsf{cost}^C(\boldsymbol{X}^{(t)}, \tilde{\boldsymbol{C}}^{(t)})$$

$$\leq \alpha \tau \mathsf{cost}^C(\boldsymbol{X}^{(t)}, \boldsymbol{C}^{(t)}) \qquad (\tilde{\boldsymbol{C}}^{(t)} \text{ is } \alpha\text{-approximation})$$

$$= \alpha \tau \sum_{i=1}^{n} d(\boldsymbol{x}_i^{(t)}, \boldsymbol{C}^{(t)})^2$$

$$\leq \alpha \tau \sum_{i=1}^{n} d(\boldsymbol{x}_i, \boldsymbol{C})^2$$

$$= \alpha \tau \mathsf{cost}^C(\boldsymbol{X}, \boldsymbol{C}).$$

Note that $|\tilde{\boldsymbol{C}}| = k$, and the above inequality holds for any $\boldsymbol{C} \in \mathcal{C}$ with $|\boldsymbol{C}| = k$. Minimizing the last item over $\boldsymbol{C} \in \mathcal{C}$ completes the proof. $\qquad\square$

Next, since we get a global constant approximation $\tilde{\boldsymbol{C}}$, we can upper bound the global sensitivities via projecting $\boldsymbol{X}$ onto $\tilde{\boldsymbol{C}}$. Concretely, the following lemma shows that $g_i^{(t)}$ (scaled by $4\tau$) is an upper bound of the global sensitivity of $\boldsymbol{x}_i$ if Assumption 5.1 holds for party $t$.

**Lemma F.4 (Upper bounding the global sensitivities for $k$-means ).** *If party $t$ of a dataset $\boldsymbol{X} \subset \mathbb{R}^d$ satisfies Assumption 5.1, then given a local $\alpha$-approximate solution $\tilde{\boldsymbol{C}}^{(t)}$, we have*

$$\sup_{\boldsymbol{C} \in \mathcal{C}} \frac{d(\boldsymbol{x}_i, \boldsymbol{C})^2}{\mathsf{cost}^C(\boldsymbol{X}, \boldsymbol{C})} \leq \frac{4\alpha\tau d(\boldsymbol{x}_i^{(t)}, \tilde{\boldsymbol{C}}^{(t)})^2}{\mathsf{cost}^C(\boldsymbol{X}^{(t)}, \tilde{\boldsymbol{C}}^{(t)})} + \frac{4\alpha\tau \sum_{j \in \boldsymbol{B}_{\pi(i)}^{(t)}} d(\boldsymbol{x}_j^{(t)}, \tilde{\boldsymbol{C}}^{(t)})^2}{|\boldsymbol{B}_{\pi(i)}^{(t)}|\mathsf{cost}^C(\boldsymbol{X}^{(t)}, \tilde{\boldsymbol{C}}^{(t)})} + \frac{8\alpha\tau}{|\boldsymbol{B}_{\pi(i)}^{(t)}|}. \qquad (1)$$

*Proof.* Let the multi-set $\pi(\boldsymbol{X}) := \{\tilde{\boldsymbol{c}}_{\pi(i)} : i \in [n]\}$ be the projection of $\boldsymbol{X}$ to $\tilde{\boldsymbol{C}}$. We denote $s_{\boldsymbol{X}}(\boldsymbol{x}_i)$ to be $\sup_{\boldsymbol{C} \in \mathcal{C}} \frac{d(\boldsymbol{x}_i, \boldsymbol{C})^2}{\mathsf{cost}^C(\boldsymbol{X}, \boldsymbol{C})}$ for $i \in [n]$. Similarly, $s_{\pi(\boldsymbol{X})}(\tilde{\boldsymbol{c}}_l) := \sup_{\boldsymbol{C} \in \mathcal{C}} \frac{d(\tilde{\boldsymbol{c}}_l, \boldsymbol{C})^2}{\mathsf{cost}^C(\pi(\boldsymbol{X}), \boldsymbol{C})}$ for $l \in [\tilde{k}]$. First we show that for any $\boldsymbol{C} \in \mathcal{G}$, the $k$-means objective of the multi-set $\pi(\boldsymbol{X})$ w.r.t. $\boldsymbol{C}$ can be upper bounded by that of $\boldsymbol{X}$ with a constant factor.

$$\mathsf{cost}^C(\pi(\boldsymbol{X}), \boldsymbol{C}) = \sum_{i=1}^{n} d(\tilde{\boldsymbol{c}}_{\pi(i)}, \boldsymbol{C})^2$$

$$= \sum_{i=1}^{n} \min_{l \in [k]} d(\tilde{\boldsymbol{c}}_{\pi(i)}, \boldsymbol{c}_l)^2$$

$$\leq \sum_{i=1}^{n} \min_{l \in [k]} \left( 2d(\boldsymbol{x}_i, \boldsymbol{c}_l)^2 + 2d(\boldsymbol{x}_i, \tilde{\boldsymbol{c}}_{\pi(i)})^2 \right) \qquad \text{(triangle inequality for } d^2)$$

$$= 2\mathsf{cost}^C(\boldsymbol{X}, \boldsymbol{C}) + 2\mathsf{cost}^C(\boldsymbol{X}, \tilde{\boldsymbol{C}})$$

$$\leq 2(1 + \alpha\tau)\mathsf{cost}^C(\boldsymbol{X}, \boldsymbol{C}) \qquad \text{(Lemma F.3)}$$

$$\leq 4\alpha\tau \mathsf{cost}^C(\boldsymbol{X}, \boldsymbol{C}). \qquad (\alpha\tau \geq 1) \qquad (2)$$

Then for any $\boldsymbol{C} \in \mathcal{C}$ and $\boldsymbol{x}_i \in \boldsymbol{X}$, we have

$$d(\boldsymbol{x}_i, \boldsymbol{C})^2$$

$$
\begin{aligned}
&= \min_{l \in [k]} d(\boldsymbol{x}_i, \boldsymbol{c}_l)^2 \\
&\leq \min_{l \in [k]} \left( 2d(\boldsymbol{x}_i, \tilde{\boldsymbol{c}}_{\pi(i)})^2 + 2d(\tilde{\boldsymbol{c}}_{\pi(i)}, \boldsymbol{c}_l)^2 \right) && \text{(triangle inequality of } d^2) \\
&= 2d(\boldsymbol{x}_i, \tilde{\boldsymbol{c}}_{\pi(i)})^2 + 2d(\tilde{\boldsymbol{c}}_{\pi(i)}, \boldsymbol{C})^2 \\
&\leq 2d(\boldsymbol{x}_i, \tilde{\boldsymbol{c}}_{\pi(i)})^2 + 2s_{\pi(\boldsymbol{X})}(\tilde{\boldsymbol{c}}_{\pi(i)}) \mathsf{cost}^C(\pi(\boldsymbol{X}), \boldsymbol{C}) && \text{(definition of } s_{\pi(\boldsymbol{X})}) \\
&\leq 2d(\boldsymbol{x}_i, \tilde{\boldsymbol{c}}_{\pi(i)})^2 + 8\alpha\tau s_{\pi(\boldsymbol{X})}(\tilde{\boldsymbol{c}}_{\pi(i)}) \mathsf{cost}^C(\boldsymbol{X}, \boldsymbol{C}) && \text{(from (2))} \\
&= 2d\left(\boldsymbol{x}_i, \frac{1}{|\boldsymbol{B}_{\pi(i)}^{(t)}|} \sum_{j \in \boldsymbol{B}_{\pi(i)}^{(t)}} \boldsymbol{x}_j\right)^2 + 8\alpha\tau s_{\pi(\boldsymbol{X})}(\tilde{\boldsymbol{c}}_{\pi(i)}) \mathsf{cost}^C(\boldsymbol{X}, \boldsymbol{C}) && \text{(definition of } \tilde{\boldsymbol{C}}) \\
&\leq \frac{2}{|\boldsymbol{B}_{\pi(i)}^{(t)}|} \sum_{j \in \boldsymbol{B}_{\pi(i)}^{(t)}} d(\boldsymbol{x}_i, \boldsymbol{x}_j)^2 + 8\alpha\tau s_{\pi(\boldsymbol{X})}(\tilde{\boldsymbol{c}}_{\pi(i)}) \mathsf{cost}^C(\boldsymbol{X}, \boldsymbol{C}) && \text{(convexity of } d^2) \\
&\leq \frac{2}{|\boldsymbol{B}_{\pi(i)}^{(t)}|} \sum_{j \in \boldsymbol{B}_{\pi(i)}^{(t)}} d(\boldsymbol{x}_i, \boldsymbol{x}_j)^2 + \frac{8\alpha\tau}{|\boldsymbol{B}_{\pi(i)}^{(t)}|} \mathsf{cost}^C(\boldsymbol{X}, \boldsymbol{C}) && (s_{\pi(\boldsymbol{X})}(\tilde{\boldsymbol{c}}_{\pi(i)}) \leq \frac{1}{|\boldsymbol{B}_{\pi(i)}^{(t)}|}) \\
&\leq \left( \frac{2}{|\boldsymbol{B}_{\pi(i)}^{(t)}|} \sum_{j \in \boldsymbol{B}_{\pi(i)}^{(t)}} \frac{d(\boldsymbol{x}_i, \boldsymbol{x}_j)^2}{\mathsf{cost}^C(\boldsymbol{X}, \boldsymbol{C})} + \frac{8\alpha\tau}{|\boldsymbol{B}_{\pi(i)}^{(t)}|} \right) \mathsf{cost}^C(\boldsymbol{X}, \boldsymbol{C}).
\end{aligned}
$$

Thus,

$$
\begin{aligned}
&\frac{d(\boldsymbol{x}_i, \boldsymbol{C})^2}{\mathsf{cost}^C(\boldsymbol{X}, \boldsymbol{C})} \\
&\leq \frac{2}{|\boldsymbol{B}_{\pi(i)}^{(t)}|} \sum_{j \in \boldsymbol{B}_{\pi(i)}^{(t)}} \frac{d(\boldsymbol{x}_i, \boldsymbol{x}_j)^2}{\mathsf{cost}^C(\boldsymbol{X}, \boldsymbol{C})} + \frac{8\alpha\tau}{|\boldsymbol{B}_{\pi(i)}^{(t)}|} \\
&\leq \frac{2\tau}{|\boldsymbol{B}_{\pi(i)}^{(t)}|} \sum_{j \in \boldsymbol{B}_{\pi(i)}^{(t)}} \frac{d(\boldsymbol{x}_i^{(t)}, \boldsymbol{x}_j^{(t)})^2}{\mathsf{cost}^C(\boldsymbol{X}, \boldsymbol{C})} + \frac{8\alpha\tau}{|\boldsymbol{B}_{\pi(i)}^{(t)}|} && \text{(Assumption 5.1)} \\
&\leq \frac{2\alpha\tau}{|\boldsymbol{B}_{\pi(i)}^{(t)}|} \sum_{j \in \boldsymbol{B}_{\pi(i)}^{(t)}} \frac{d(\boldsymbol{x}_i^{(t)}, \boldsymbol{x}_j^{(t)})^2}{\mathsf{cost}^C(\boldsymbol{X}^{(t)}, \tilde{\boldsymbol{C}}^{(t)})} + \frac{8\alpha\tau}{|\boldsymbol{B}_{\pi(i)}^{(t)}|} && \text{(Lemma F.3)} \\
&\leq \frac{4\alpha\tau}{|\boldsymbol{B}_{\pi(i)}^{(t)}|} \sum_{j \in \boldsymbol{B}_{\pi(i)}^{(t)}} \frac{d(\boldsymbol{x}_i^{(t)}, \tilde{\boldsymbol{c}}_{\pi(i)})^2 + d(\boldsymbol{x}_j^{(t)}, \tilde{\boldsymbol{c}}_{\pi(i)})^2}{\mathsf{cost}^C(\boldsymbol{X}^{(t)}, \tilde{\boldsymbol{C}}^{(t)})} + \frac{8\alpha\tau}{|\boldsymbol{B}_{\pi(i)}^{(t)}|} && \text{(triangle inequality of } d^2) \\
&= \frac{4\alpha\tau d(\boldsymbol{x}_i^{(t)}, \tilde{\boldsymbol{C}}^{(t)})^2}{\mathsf{cost}^C(\boldsymbol{X}^{(t)}, \tilde{\boldsymbol{C}}^{(t)})} + \frac{4\alpha\tau \sum_{j \in \boldsymbol{B}_{\pi(i)}^{(t)}} d(\boldsymbol{x}_j^{(t)}, \tilde{\boldsymbol{C}}^{(t)})^2}{|\boldsymbol{B}_{\pi(i)}^{(t)}| \mathsf{cost}^C(\boldsymbol{X}^{(t)}, \tilde{\boldsymbol{C}}^{(t)})} + \frac{8\alpha\tau}{|\boldsymbol{B}_{\pi(i)}^{(t)}|},
\end{aligned}
$$

taking supremum over $\boldsymbol{C} \in \mathcal{C}$ completes the proof. $\qquad\square$

Now we are ready to prove Lemma F.2.

*Proof of Lemma F.2.* By Lemma F.4, since some party $t \in [T]$ satisfies Assumption 5.1, then

$$
\sup_{\boldsymbol{C} \in \mathcal{C}} \frac{\mathsf{cost}_i^C(\boldsymbol{X}, \boldsymbol{C})}{\mathsf{cost}^C(\boldsymbol{X}, \boldsymbol{C})} \leq 4\tau g_i^{(t)} \leq 4\tau \sum_{j \in [T]} g_i^{(j)},
$$

where $g_i^{(t)}$ is defined as the right side of (1) for any $t \in [T]$. Moreover,

$$
\mathcal{G} = \sum_{i \in [n]} \sum_{j \in [T]} g_i^{(j)}
$$

$$
= \sum_{j \in [T]} \sum_{i \in [n]} \left( \frac{\alpha d(\boldsymbol{x}_i^{(j)}, \tilde{\boldsymbol{C}}^{(j)})^2}{\mathsf{cost}^C(\boldsymbol{X}^{(j)}, \tilde{\boldsymbol{C}}^{(j)})} + \frac{\alpha \sum_{i' \in \boldsymbol{B}_{\pi(i)}^{(j)}} d(\boldsymbol{x}_{i'}^{(j)}, \tilde{\boldsymbol{C}}^{(j)})^2}{|\boldsymbol{B}_{\pi(i)}^{(j)}| \mathsf{cost}^C(\boldsymbol{X}^{(j)}, \tilde{\boldsymbol{C}}^{(j)})} + \frac{2\alpha}{|\boldsymbol{B}_{\pi(i)}^{(j)}|} \right)
$$

$$
= \sum_{j \in [T]} (\alpha + \alpha + 2k\alpha)
$$

$$
= 2(k+1)\alpha T.
$$

Hence, $\zeta = O(\tau)$ and $\mathcal{G} = O(\alpha k T)$, which completes the proof. $\qquad \square$

# G  Robust Coresets for VRLR and VKMC

In this section, we prove that even if the data assumptions 4.1 and 5.1 fail to hold, Algorithms 2 and 3 still provide robust coresets for VRLR (Theorem G.3) and VKMC (Theorem G.4) in the flavor of approximating with *outliers*.

## G.1  Robust coreset

In this section, we introduce a general definition of robust coreset. For preparation, we first give some notations for a function space, which can be easily specialized to the cases for VRLR and VKMC. Given a dataset $\boldsymbol{X}$ of size $n$, let $F$ be a set of cost functions from $\boldsymbol{X}$ to $\mathbb{R}_{\geq 0}$. For a subset $S \subseteq [n]$ with a weight function $w : S \to \mathbb{R}_{\geq 0}$, we denote $f(S)$ to be the weighted total cost over $S$ for any $f \in F$, i.e., $f(S) = \sum_{i \in S} w(i) f(\boldsymbol{x}_i)$. With a slight abuse of notation, we can see $\boldsymbol{X}$ as $[n]$ with unit weight such that $f(\boldsymbol{X}) = \sum_{i \in [n]} f(\boldsymbol{x}_i)$. Now we define the *robust coreset* as follows.

**Definition G.1 (Robust coreset).** Let $\beta \in [0, 1)$, and $\varepsilon \in (0, 1)$. Given a set $F$ of functions from $\boldsymbol{X}$ to $\mathbb{R}_{\geq 0}$, we say that a weighted subset of $S \subseteq [n]$ is a $(\beta, \varepsilon)$-robust coreset of $\boldsymbol{X}$ if for any $f \in F$, there exists a subset $O_f \subseteq [n]$ such that

$$
\frac{|O_f|}{n} \leq \beta, \frac{|S \cap O_f|}{|S|} \leq \beta,
$$

$$
|f(\boldsymbol{X} \backslash O_f) - f(S \backslash O_f)| \leq \varepsilon f(\boldsymbol{X}).
$$

Roughly speaking, we allow a small portion of data to be treated as outliers and neglected both in $\boldsymbol{X}$ and $S$ when considering the quality of $S$. Note that a $(0, \varepsilon)$-robust coreset is equivalent to a standard $\varepsilon$-coreset, and $S$ provides a slightly weaker approximation guarantee with additive error if $\beta > 0$. Also note that our definition of robust coreset is a bit different from that in previous work [22, 32, 69], which focus on generating robust coresets from uniform sampling, but basically they all capture similar ideas. This is because we will be interested in the robustness of importance sampling under the case where a small percentage of data have unbounded sensitivity gap in Algorithm 1, and the above definition gives simpler results.

We propose the following theorem to show that $(S, w)$ returned by Algorithm 1 is a $(\beta, \varepsilon)$-robust coreset when size $m$ is large enough.

**Theorem G.2 (The robustness of Algorighm 1).** *Let $\beta, \varepsilon \in (0, 1)$. Given a dataset $\boldsymbol{X}$ of size $n$ and a set $F$ of functions from $\boldsymbol{X}$ to $\mathbb{R}_{\geq 0}$, let $g_i = \sum_{j \in [T]} g_i^{(j)}$ and $\mathcal{G} = \sum_{i \in [n]} g_i$. Let $S \subseteq [n]$ be a sample of size $m$ drawn i.i.d from $[n]$ with probability proportional to $\{g_i : i \in [n]\}$, where each sample $i \in [n]$ is selected with probability $\frac{g_i}{\mathcal{G}}$ and has weight $w(i) := \frac{\mathcal{G}}{m g_i}$. If $\forall i \in [n]$, $j \in [T]$ we have $g_i^{(j)} \geq 1/n$, let $s_i := \sup_{f \in F} \frac{f(\boldsymbol{x}_i)}{f(\boldsymbol{X})}$ and $c = \frac{2 \sum_{i \in [n]} s_i}{\beta T}$. If*

$$
m = O \left( \frac{c^2 \mathcal{G}^2}{\varepsilon^2} \left( \dim(F) + \log \frac{1}{\delta} \right) \right), \tag{3}
$$

*where $\dim(F)$ is the pseudo-demension of $F$. Then with probability $1 - \delta$, $(S, w)$ is a $(\beta, \varepsilon)$-robust coreset of $\boldsymbol{X}$.*

The proof is in Section G.4. Recall that the term $\frac{s_i}{g_i}$ represents the sensitivity gap of point $\boldsymbol{x}_i$, and Algorithm 1 guarantees sublinear communication complexity only if the maximum sensitivity gap $\zeta$ over all points is independent of $n$. The main idea in the above theorem is that we can reduce the portion of potential outliers (with large sensitivity gap) to a small constant both in $\boldsymbol{X}$ and $S$ via scaling sample size $m$ by a sufficiently large constant.

## G.2 Robust coresets for VRLR

The following theorem shows that Algorithm 2 returns a robust coreset for VRLR when sample size $m$ is large enough. Note that $m$ is still independent of $n$.

**Theorem G.3** (**Robust coresets for VRLR**). *For a given dataset $\boldsymbol{X} \subset \mathbb{R}^d$, integer $T \geq 1$ and constants $\beta, \varepsilon, \delta \in (0,1)$, with probability at least $1 - \delta$, Algorithm 2 constructs a $(\beta, \varepsilon)$-robust coreset for VRLR of size*

$$m = O\left(\frac{d^4}{\varepsilon^2 \beta^2 T^2}\left(d^2 + \log\frac{1}{\delta}\right)\right),$$

*and uses communication complexity $O(mT)$.*

*Proof.* By Theorem G.2, in VRLR, $F = \{f_{\boldsymbol{\theta}} : f_{\boldsymbol{\theta}}(\boldsymbol{x}) = (\boldsymbol{x}^\top \boldsymbol{\theta} - \boldsymbol{y})^2 + R(\boldsymbol{\theta})/n, \boldsymbol{\theta} \in \mathbb{R}^d\}$. Note that in Theorem 4.2, $g_i^{(j)} = \|\boldsymbol{u}_i^{(j)}\|^2 + \frac{1}{n} \geq \frac{1}{n}$, $\mathcal{G} = O(d)$ and $\sum_{i \in [n]} s_i = O(d)$, we have $c\mathcal{G} = O(\frac{d^2}{\beta T})$. Plugging $c\mathcal{G} = O(\frac{d^2}{\beta T})$ and $\dim(F) = d^2$ into (3) completes the proof. $\qquad\square$

## G.3 Robust coresets for VKMC

The following theorem shows that Algorithm 3 returns a robust coreset for VKMC when sample size $m$ is large enough. Note that $m$ is still independent of $n$.

**Theorem G.4** (**Robust coresets for VKMC**). *For a given dataset $\boldsymbol{X} \subset \mathbb{R}^d$, an $\alpha$-approximation algorithm for $k$-means with $\alpha = O(1)$, integers $k \geq 1$, $T \geq 1$ and constants $\beta, \varepsilon, \delta \in (0,1)$, with probability at least $1 - \delta$, Algorithm 3 constructs a $(\beta, \varepsilon)$-robust coreset for VKMC of size*

$$m = O\left(\frac{\alpha^2 k^4}{\varepsilon^2 \beta^2}\left(dk + \log\frac{1}{\delta}\right)\right),$$

*and uses communication complexity $O(mT)$.*

*Proof.* By Theorem G.2, in VKMC, $F = \{f_{\boldsymbol{C}} : f_{\boldsymbol{C}}(\boldsymbol{x}) = d(\boldsymbol{x}, \boldsymbol{C})^2 = \min_{\boldsymbol{c} \in \boldsymbol{C}} d(\boldsymbol{x}, \boldsymbol{c})^2, \boldsymbol{C} \in \mathcal{C}, |\boldsymbol{C}| = k\}$. Note that in Theorem 5.2, $g_i^{(j)} \geq \frac{1}{n}$, $\mathcal{G} = O(\alpha k T)$ and $\sum_{i \in [n]} s_i = O(k)$, we have $c\mathcal{G} = O(\frac{\alpha k^2}{\beta})$. Plugging $c\mathcal{G} = O(\frac{d^2}{\beta T})$ and $\dim(F) = dk$ into (3) completes the proof. $\qquad\square$

## G.4 Proof of Theorem G.2

We first introduce the following lemma which mainly shows that importance sampling generates an $\varepsilon$-approximation of $\boldsymbol{X}$ on the corresponding weighted function space.

**Lemma G.1** (**Importance sampling on a function space [2, 22]**). *Given a set $F$ of functions from $\boldsymbol{X}$ to $\mathbb{R}_{\geq 0}$ and a constant $\varepsilon \in (0,1)$, let $S$ be a sample of size $m$ drawn i.i.d from $[n]$ with probability proportional to $\{g_i : i \in [n]\}$. If $g_i = \Omega(\frac{1}{n})$ for any $i \in [n]$, and let $\mathcal{G} = \sum_{i \in [n]} g_i$. If*

$$m = O\left(\frac{1}{\varepsilon^2}\left(\dim(F) + \log\frac{1}{\delta}\right)\right),$$

*where $\dim(F)$ is the pseudo-demension of $F$. Then with probability $1 - \delta$, $\forall f \in F$ and $\forall r \geq 0$,*

$$\left| \sum_{i \in [n], \frac{f(\boldsymbol{x}_i)}{g_i} \leq r} f(\boldsymbol{x}_i) - \sum_{i \in S, \frac{f(\boldsymbol{x}_i)}{g_i} \leq r} \frac{\mathcal{G}}{m g_i} f(\boldsymbol{x}_i) \right| \leq \mathcal{G}\varepsilon r.$$

Now we are ready to prove Theorem G.2.

*Proof of Theorem G.2.* Recall that $c = \frac{2 \sum_{i \in [n]} s_i}{\beta T}$. Let $O \subseteq [n]$ be defined as

$$O := \{i \in [n] : s_i \geq cg_i\}.$$

Note that $g_i = \sum_{j \in [T]} g_i^{(j)} \geq \frac{T}{n}$, and $\sum_{i \in [n]} s_i \geq \sum_{i \in O} s_i \geq \sum_{i \in O} cg_i \geq |O| \cdot \frac{cT}{n}$. Hence,

$$\frac{|O|}{n} \leq \frac{\sum_{i \in [n]} s_i}{cT} = \frac{\beta}{2} < \beta. \tag{4}$$

Let $p$ be the probability that a point in $S$ belongs to $O$, then

$$p = \frac{\sum_{i \in O} g_i}{\sum_{i \in [n]} g_i} \leq \frac{\sum_{i \in O} s_i}{c \sum_{i \in [n]} g_i} \leq \frac{\sum_{i \in O} s_i}{cT} \leq \frac{\sum_{i \in [n]} s_i}{cT} = \frac{\beta}{2}.$$

Hence, by a standard multiplicative Chernoff bound, if $m = \Omega(\frac{1}{\beta} \log 1/\delta)$, then with probability $1 - \delta/2$, we have

$$\frac{|S \cap O|}{|S|} \leq \beta. \tag{5}$$

For any $f \in F$, we define a subset $O_f \subseteq O$ as follows,

$$O_f := \{i \in [n] : \frac{f(\boldsymbol{x}_i)}{f(\boldsymbol{X})} \geq cg_i\}.$$

By (4) and (5), we have that $\frac{|O_f|}{n} \leq \beta$ and $\frac{|S \cap O_f|}{|S|} \leq \beta$. Note that $f(\boldsymbol{x}_i)/g_i \geq cf(\boldsymbol{X})$ if and only if $i \in O_f$. Let $r = cf(\boldsymbol{X})$ and plug it into Lemma G.1, then

$$\left| \sum_{i \in [n], \frac{f(\boldsymbol{x}_i)}{g_i} \leq r} f(\boldsymbol{x}_i) - \sum_{i \in S, \frac{f(\boldsymbol{x}_i)}{g_i} \leq r} \frac{\mathcal{G}}{mg_i} f(\boldsymbol{x}_i) \right| = |f(\boldsymbol{X} \setminus O_f) - f(S \setminus O_f)| \leq \mathcal{G}c\varepsilon f(\boldsymbol{X}),$$

scaling $\varepsilon$ by $\frac{1}{c\mathcal{G}}$ completes the proof. $\qquad \square$