# OpenReview forum: "Coresets for Vertical Federated Learning: Regularized Linear Regression and $K$-Means Clustering"
_NeurIPS.cc/2022/Conference — NeurIPS 2022 Accept_

### Official Review · Reviewer_XK2T · 2022-07-06

**Rating:** 7
**Confidence:** 4
**Soundness:** 3 good
**Presentation:** 3 good
**Contribution:** 3 good

**Summary:**

This paper forges a link between vertical federated learning where the attributes of the data are spread along multiple machines, to coresets - a data summarization that approximates the cost of every query on the data.
Specifically speaking, they focus on the problems of k-means and least mean squares regression.

The main idea is to reduce the communication complexity of an algorithm by computing a coreset for the distributed data - to do so, the authors extend the framework of importance (sensitivity) sampling to the vertical federated learning settings; see Algorithm 1 and Theorem 3.1.

Focusing on Least squares regression the authors demonstrate that it is often challenging to build a "strong" (for every query) coreset for VRLR - the lower bound on the communication complexity is Ω(n).
Then, they show how under a loose assumption, this can be done efficiently. According to this presumption, the subspace produced by data on any one party should not be entirely contained inside the subspace produced by other parties. Under such an assumption, one can get a coreset of size-independent from n, polynomial in the dimension d, and the appropriation error.

The same scenario hold for k-means - which requires a communication complexity Ω(n). However, assuming any two data points that may be differentiated can also be differed on a party that is "important" to some amount.

Finally, experimental results show that coresets reduce the communication complexity and maintain a good quality solution.


**Questions:**



Can you please explain (intuitively) the relation between $\gamma$ in Theorem 4.2 to the parameter which represents the maximum sensitivity gap overall points. Same for \tau in Theorem 5.2.


Can you give please more details about the parameter which represents the maximum sensitivity gap, how is it related to the data e.g., in linear regression?

**Limitations:**

The coreset requires a specific assumption on the data (that may not be satisfied in real-world).

**Strengths And Weaknesses:**

I like the paper, mainly, the idea of using coreset in this new realm. The ideas are very interesting, and the findings are robust.



Some comments and questions:
1. Theorem 3.1 - S, W is not defined in the theorem.
2. The empirical results are good but can be extended, e.g., 1. try on a larger number of parties, other datasets, and more.
3. Please define (or at least explain) a robust coreset in the manuscript.

---

> ### Author Response · Authors · 2022-08-02
> **Response to Reviewer XK2T**
>
> Dear Reviewer XK2T,
>
> Thank you for your positive review and feedback. We answer your specific questions below.
>
> > ''Can you please explain (intuitively) the relation between Theorem 4.2 to the parameter which represents the maximum sensitivity gap overall points. Same for $\tau$ in Theorem 5.2. Can you give please more details about the parameter which represents the maximum sensitivity gap, how is it related to the data e.g., in linear regression?''
>
> For VRLR in theorem 4.2, as the parameter $\gamma$ becomes larger, the sensitivity gap $\zeta$ becomes smaller (Lemma E.2). Intuitively, $\gamma\in (0,1]$ represents the degree of orthonormal among data in different parties. As the larger $\gamma$ is, the more orthonormal among the column spaces of $X^{(j)}$, and thus $U$ is more close to the orthonormal basis computed on $X$ directly. Consequently, we have
> $$\sup\limits_{\theta\in \mathbb R^d}\frac{cost^R_i(X,\theta)}{cost^R(X,\theta)}\approx \sum_{j\in [T]} \sup\limits_{\theta\in R^d}\frac{cost^R_i(X^{(j)},\theta)}{cost^R(X^{(j)},\theta)},$$
> which implies that the sensitivity gap $\zeta$ is more close to 1 if each $g_i^{(j)}\approx \sup\limits_{\theta\in R^d}\frac{cost^R_i(X^{(j)},\theta)}{cost^R(X^{(j)},\theta)}$ is close to the local sensitivity.
>
> For VKCM in Theorem 5.2, as the parameter $\tau$ becomes smaller, the sensitivity gap $\zeta$ becomes smaller (Lemma F.2). Intuitively, as $\tau$ is more close to 1, Assumption 5.1 implies that there exists a party $t\in [T]$ whose local pairwise distances $\|x_i^{(t)} - x_j^{(t)}\|$s are close to the corresponding global pairwise distances $\|x_i - x_j\|$s. Then the global sensitivity is approximately dominated by the local sensitivity $g_i^{(t)}$, which implies that $g_i=\sum_{t\in [T]} g_i^{(t)}$ is indeed an approximate upper bound of the global sensitivity.
>
> We will add more discussions in the future version.

---

> > ### Comment · Reviewer_XK2T · 2022-08-08
> > **Thank you for addressing my questions.**
> >
> > I have read the author's rebuttal and I keep my score.

---

> > > ### Author Response · Authors · 2022-08-08
> > > **Reply to to Reviewer XK2T**
> > >
> > > Thanks a lot for your support!

---

### Official Review · Reviewer_4KgV · 2022-07-10

**Rating:** 6
**Confidence:** 4
**Soundness:** 3 good
**Presentation:** 2 fair
**Contribution:** 3 good

**Summary:**

The paper gives a unified framework to construct coresets for the regularized regression and k-means clustering  in the vertical federated learning setting where the features of the data are distributed over multiple parties. The aim is to make the communication complexity sublinear in the no. of data points. The high level idea is to calculate local sensitivities for points at each party and then use them to sample points at each party. Finally take a union of these samples. The main contribution is to show that this intuitive technique gives a coreset with much lower communication complexity  under some assumptions on the data. When there are no assumptions, the authors also give some lower bounds. The authors also validate their theoretical claims with empirical results on real world dataset.

**Questions:**

Please clarify the following questions
1) Will the results for regularized regression also extend to the setting when the data matrix is low rank? Is there any assumption here that data is full rank? Also the theoretical results hold for linear regression( no regularization) also? you have already shown some empirical results in appendix.
2) Instead of giving the calculated values of communication complexity in the graphs, is it possible to compute time required for the communication in the experimental setup?


**Limitations:**

Please see the weaknesses/ questions section. Also please do a complete proof reading and try to make some sentence constructions better.

**Strengths And Weaknesses:**

Strengths:
1) The use of coresets in the VFL setting to reduce communication complexity, to the best of my knowledge is novel. It is an important and interesting application.
2) The idea is simple and intuitive
3) The theory and the proofs appear sound and correct. I didnot go through the proofs in the appendix in details but had a quick look.
4) Empirical evaluations validate theoretical claims in terms of the cost.

Weaknesses:
1)The writing is sloppy at a few places. Blelow are a few examples:
        i)Line 22: "Most of the VFL literature focus on the privacy issue, and design secure..." should be and designing.
        ii) Line 66 "how to defend again" should be against.
        iii) Line 68 last sentence is not proper.
        iv)Algorithm 1, line 3 summation should be over j.
2) I could not find the details of the hardware system used for the implementation. Was it really a distributed set up or was the data just broken into parts and experiments were done on single system?
3) Most of the proof techniques(other than lower bounds) and the coreset construction techniques  are very well known in literature and there is lack of novelty in that sense.
4) The remarks regarding robust coresets are given without even the definiton of robust coreset in the main paper. I believe atleast the definition and some intuition/discussion as to why the robust coresets are possible without assumptions should be in main body otherwise the main paper becomes less readable.

---

> ### Author Response · Authors · 2022-08-02
> **Response to Reviewer 4KgV**
>
> Dear Reviewer 4KgV,
>
> Thanks for your time and effort! We have fixed typos according to your suggestions and will move more contents of robust coreset to the main text in the future version.
>
> > ''I could not find the details of the hardware system used for the implementation. Was it really a distributed set up or was the data just broken into parts and experiments were done on single system?''
>
> We conduct experiments on a single system that simulates the distributed settings and reports the training loss/test loss/communication complexity.
>
> > ''Will the results for regularized regression also extend to the setting when the data matrix is low rank? Is there any assumption here that data is full rank?''
>
> Assumption 4.1 does not require each local matrix $X^{(j)}$ to be full rank. For each party $j$, the number $d_j'$ of the orthonormal basis $U^{(j)}$ might be smaller than the number of features $d_j$, which implies that the matrix $X^{(j)}$ can be low rank. However, we do need some kind of ''full rank'' assumptions between parties, i.e., the combination $U$ of $U^{(j)}$s should be ''full rank''.
>
> > ''Also the theoretical results hold for linear regression (no regularization) also?''
>
> For linear regression without a regularizer, our theorems/results also hold.
>
> > ''Instead of giving the calculated values of communication complexity in the graphs, is it possible to compute the time required for the communication in the experimental setup?''
>
> Since we use a single system to simulate the distributed setting, it may be inaccurate to count the communication time. Thus, we report the communication complexity that also matches our theoretical results.

---

### Official Review · Reviewer_Nzai · 2022-07-11

**Rating:** 7
**Confidence:** 3
**Soundness:** 3 good
**Presentation:** 3 good
**Contribution:** 3 good

**Summary:**

This work considers how to subsample the datasets via coresets under the vertical federated learning (VFL) setting. By doing so, the communication cost of VFL algorithms can be sharply reduced. The authors first introduced a unified coreset construction algorithm in VFL, which requests an importance value for each sample, and then proposed how to estimate the importance values for regularized linear regression and k-means clustering, respectively. Experimental results show that the proposed algorithms can reduce the communication complexity without harming the model performance significantly.

**Questions:**

## Definition
How is the cost evaluated in Section 6? In Definition 2.3 and Definition 2.4, the cost is evaluated on the coresets (i.e. $\sum_{I \in S}$). However, in my opinion, although the models are trained/computed on the coresets, the cost should be evaluated on the entire dataset (i.e., $\sum_{I \in \[n\]}$) when assessing the performance.

## Privacy analysis
I suggest the authors discuss the privacy of the proposed algorithms. To be specific, analyze what can be leaked from the messages transmitted in Algorithm 1. I also suggest the authors clarify the security assumption. It seems the server and clients are assumed to be semi-honest. But when they become malicious, is the proposed algorithm safe?

## Experiments
How does the CENTRAL baseline solve the ridge regression problem? Does it solve by the closed-form solution (i.e., $(X^TX + \lambda I)^{-1} X^T y$) or gradient descent? If by the closed-form solution, how does it solve the Lasso regression and elastic nets in Appendix A.2?

In Table 1, why the communication complexity of C-SAGA decreases w.r.t. the size of coresets? As indicated by Theorem 2.5, the communication complexity should be $\Omega(mT)$ where $m$ is related to the size of coresets.

Moreover, I suggest the authors report the communication of coreset construction and model training, respectively. It would help readers understand the overhead of your method.

## Some other points
- The caption of Figure 1 is missing.
- The term “robust coreset” is not introduced.
- In many previous VFL studies, there is no such a central server. However, it seems the central server can be substituted by Party T in Algorithm 1, so it looks good to me.


**Limitations:**

The authors have discussed future directions in Section 7.

**Strengths And Weaknesses:**

## Strengths
Overall, this is a solid work for me. The techniques are well introduced and the paper is easy to follow. Although the authors made two ideal assumptions when proposing algorithms for regularized linear regression and k-means clustering, they also provided the theoretical analysis when the assumptions are not satisfied. The experimental results are positive: the proposed algorithms work well when using less than 4% of the training data.
## Weakness
Privacy is an important issue in VFL, but the authors did not analyze the privacy of the proposed algorithms. And the experimental results can be explained more in-depth.

---

> ### Author Response · Authors · 2022-08-02
> **Response to Reviewer Nzai**
>
> Dear Reviewer Nzai,
>
> Thanks for your time and effort! We will revise our paper according to your suggestions by moving more contents of robust coreset to the main text and adding more discussions on privacy analysis.
>
> ## Definition
>
> > ''How is the cost evaluated in Section 6? In Definition 2.3 and Definition 2.4, the cost is evaluated on the coresets (i.e. $\sum_{i\in S}$). However, in my opinion, although the models are trained/computed on the coresets, the cost should be evaluated on the entire dataset (i.e., $\sum_{i\in [n]}$) when assessing the performance.''
>
> In Section 6, we partition the YearPredictionMSD dataset into a training set and a testing set. For the ridge regression task, we train it using the coreset/uniformly sampled subset from the training set to learn a model $\theta$, and report the regression loss $cost^R(T, \theta)$ on the testing set $T$. For the $k$-means clustering task, because it is an unsupervised learning problem, we optimize on the training set and get the model $C$, then report the clustering cost $cost^C(X, C)$ on the full training set $X$.
>
> We make more clarification in the revised PDF version.
>
> ## Privacy Analysis
>
> > ''I suggest the authors discuss the privacy of the proposed algorithms. To be specific, analyze what can be leaked from the messages transmitted in Algorithm 1. I also suggest the authors clarify the security assumption. It seems the server and clients are assumed to be semi-honest. But when they become malicious, is the proposed algorithm safe?''
>
> We agree that privacy is very important in federated learning and will add more discussions later. The whole framework can be decomposed into two parts: the coreset construction and the model training.
>
> As for the coreset construction part (Algorithm 1), the privacy leakage comes from the "sensitivity score" $g_i^{(j)}$ of the data points in different parties. To tackle this problem, we can use multi-party computation/secure aggregation such as [Bonawtiz et al, Practical Secure Aggregation for Privacy-Preserving Machine Learning, CCS'17] to transport the sum $g_i=\sum_{j=1}^T g_i^{(j)}$ to the server without letting the server know the exact values of $g_i^{(j)}$s (Line 7 of Algorithm 1). By applying secure aggregation, the server only knows $(S,w)$ and $\mathcal{G}^{(j)}$s. We added a comment in the revised PDF to illustrate this (footnote, page 5).
>
> For the model training part, we can apply the secure VFL algorithms if existed, e.g., using homomorphic encryption on SAGA for regression (it is an extension from SGD to SAGA [27]). For clustering in the VFL setting, currently we do not know any secure methods.
>
> The previous discussion assumes the semi-honesty model. Suppose some parties are malicious. For instance, party $j$ can report a large enough $\mathcal{G}^{(j)}$ (Line 2 of Algorithm 1) such that the server sets the number of samples $a_j\approx m$ in party $j$ (Line 4 of Algorithm 1), where $m$ is the coreset size of $S$. Consequently, party $j$ can sample a large multiset $S^{(j)}$ which heavily affects the resulting coreset $S$. By e.g., reporting $S^{(j)}$ of uniform samples, party $j$ can make $S$ close to uniform sampling and loss the theoretical guarantees in Theorem 3.1.
>
>
> ## Experiment
>
> > ''How does the CENTRAL baseline solve the ridge regression problem? Does it solve by the closed-form solution or gradient descent? If by the closed-form solution, how does it solve the Lasso regression and elastic nets in Appendix A.2?''
>
> In the experiment, we use the scikit-learn package to solve the problems. As for the ridge regression, scikit-learn uses the lbfgs solver (some second-order methods). For lasso and elastic net, scikit-learn uses the SAGA solver.
>
> > ''In Table 1, why the communication complexity of C-SAGA decreases w.r.t. the size of coresets? As indicated by Theorem 2.5, the communication complexity should be $\Omega(mT)$ where $m$ is related to the size of coresets.''
>
> We are sorry that we forgot to multiple the coreset size when computing the communication complexity of SAGA in Table 1. A larger coreset size indeed has a larger communication complexity. Table 1 is fixed and the graph in the main content is updated in the revised pdf.
>
> > ''Moreover, I suggest the authors report the communication of coreset construction and model training, respectively. It would help readers understand the overhead of your method.''
>
> Thanks for your suggestion. We will report the communication complexity of coreset construction and model training in the future version. In general, the communication complexity of coreset construction is very small compared to that of model training.

---

> > ### Comment · Reviewer_Nzai · 2022-08-08
> > **Re Response to Reviewer Nzai**
> >
> > Thanks for the response, which addresses my concerns and questions. I will maintain my scores.
> >
> > BTW, it would be better if the authors could describe the security model in the revised paper. Besides, add a discussion that the central server can be replaced with any party (e.g. Party $T$), since in practical VFL applications, there is usually no such a central server.

---

> > > ### Author Response · Authors · 2022-08-08
> > > **Re Re Response to Reviewer Nzai**
> > >
> > > Thanks a lot for your support! We will address your comments in the updated version.

---

### Meta-Review · Area_Chair_vr8m · 2022-08-30

**Recommendation:** Accept
**Confidence:** Less certain

**Metareview:**

The reviewers have converged around the idea that the paper proposes an interesting approach to vertical federated learning; they also conclude that the authors have provided replies to reviews that answered questions and provided useful clarifications, which encourages the acceptance of the paper.

I will stress the need for the authors to properly include further updates for the camera ready version of the paper; in their reply, they indeed make several promises to reviewers and it is important that such updates be properly included (last comment to NZai, intermediary comment to XK2T).

**Award:**

No

---

### Decision · Program_Chairs · 2022-09-14

Accept